# QKV Projections Require a Fraction of Their Memory

**Malik Khalaf, Yara Shamshoum, Nitzan Hodos, Yuval Sieradzki, Assaf Schuster**
Department of Computer Science
Technion, Israel Institute of Technology
{`malikkhalaf,yara-sh,hodosnitzan,syuvsier`}@campus.technion.ac.il

## Abstract

The Multi-Head Attention mechanism is central to LLM operation, and multiple works target its compute and memory efficiency during training. While most works focus on approximating the scaled dot product, the memory consumption of the linear projections that compute the $Q$, $K$, and $V$ tensors from the input $x$ is often overlooked. To address this, we propose Point-Approximate Matrix Multiplication (PAMM), a novel tensor compression technique that compresses the activations of the $Q, K, V$ projections in attention layers by a factor of up to $\times 512$, effectively erasing their memory footprint, while achieving similar or better final perplexity. PAMM is fully composable with efficient attention techniques such as FlashAttention, making it a practical and complementary method for memory-efficient LLM training. Our code is publicly available at `https://gitlab.com/MalikKhalaf4/pamm`.

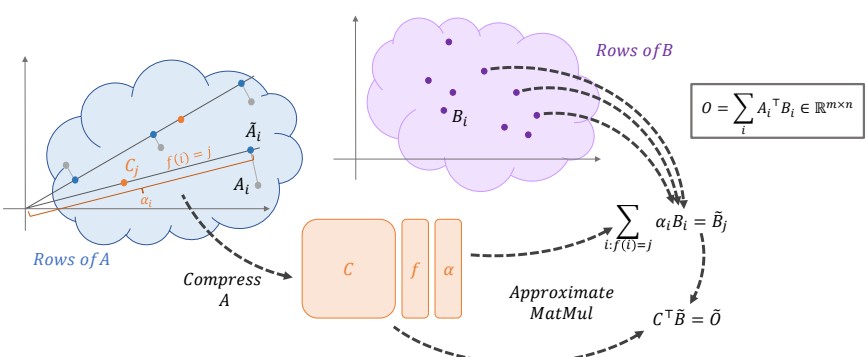

Figure 1: **Illustration of Point-Approximate Matrix Multiplication (PAMM)**. PAMM approximates the matrix multiplication $O = A^\top B$ for $A \in \mathbb{R}^{b \times n}$ and $B \in \mathbb{R}^{b \times m}$ in two stages. First, each row $A_i$ is represented by $\tilde{A}_i$, defined as the closest point to $A_i$ on the line spanned by the best representative $C_j$. Instead of storing the full matrix $A$, PAMM keeps only a small set of $k$ generators $C \in \mathbb{R}^{k \times n}$, together with an assignment mapping $f \in \mathbb{R}^b$ and scaling factors $\alpha \in \mathbb{R}^b$. Using these, PAMM approximates the product by first contracting $B$ into $\tilde{B} \in \mathbb{R}^{k \times m}$, and then calculating $\tilde{O} = C^\top \tilde{B}$ (equivalently, $\tilde{O} = \tilde{A}^\top B$). In our setting, PAMM is applied to approximate the gradient of the projection matrices $Q, K, V$ during backpropagation, e.g. $\widetilde{\nabla W}_Q = \tilde{X}^\top \nabla Q$. Since the number of generating points $\{C_j\}_{j=1}^k$ is typically very small, the memory required to store $X$ is drastically reduced.

## 1 Introduction

Large Language Models (LLMs) have become the foundation of modern NLP research, achieving state-of-the-art results across a wide range of benchmarks (Liu et al., 2019; Brown et al., 2020; et al., 2024a). However, as model sizes continue to increase, training them becomes increasingly

challenging, especially when considering limited resources in terms of compute and memory. As a result, numerous techniques have been developed to improve memory efficiency during training.

Since most LLMs are transformer-based models (Vaswani et al., 2017), reducing the memory cost of the attention mechanism has been researched extensively, especially during training (Wang et al., 2020; Kitaev et al., 2020; Choromanski et al., 2022). These works often discuss extending context length, training with larger batch sizes, or reducing the time and memory complexity of the attention operation itself. However, when looking at the overall GPU memory consumption of the attention layer, a component that is often overlooked becomes prominent: the memory requirement of the linear projection layers, which project the input $x$ into the $Q, K, V$ tensors used by scaled dot product attention. Since the input $x$ is required for backward computation, it is saved by each layer during the forward pass, quickly accumulating to contribute as much as 20% of the total peak GPU memory required by the attention blocks.

High memory consumption by intermediate activations during training is not unique to attention projections. Recent work Shamshoum et al. (2025) shows that activation memory far exceeds that of parameters or optimizer states in practical settings, since activations are the only component that scales with batch size and sequence length. This contrasts with the recent trend of focusing on optimizer state compression to reduce memory consumption in LLMs (Zhao et al., 2024; Hao et al., 2024; Muhamed et al., 2024; Zhang et al., 2024)."

These recent compression techniques use low-rank approaches that exploit redundancies in the hidden dimension. In this work, we identify a much more substantial source of redundancy: the sequence dimension. We point out that activation tensors in transformer models are often highly redundant across tokens, due to repeated patterns, padding, or local contextual similarity (Dao et al., 2022; Dao, 2023; Krell et al., 2022; Dai et al., 2020; Yao et al., 2022). This redundancy is expected, as the dimensionality of each token representation is much smaller than the number of token embeddings in the activation tensor. We discuss this point in Section 3.1.

To exploit this redundancy, we propose Point-Approximate Matrix Multiplication (**PAMM**), a simple yet effective tensor compression method that stores only a small, representative subset of token activations to improve memory efficiency. The omitted tokens are approximated using scaled versions of the stored subset. This process can be viewed as a form of approximate clustering: rather than storing the full activations, we only save representative points. However, clustering token embeddings on-the-fly is computationally prohibitive, as clustering algorithms' complexity is often quadratic in dataset size or worse (Jin & Han, 2010). We show that, surprisingly, selecting representative tokens at random is sufficient.

In this work, we apply PAMM to the $Q, K, V$ projections of the attention block, motivated by recent works that show the activations of the attention layers in an LLM exhibit clustering behavior (Geshkovski et al., 2024) and that the attention layer is much more amenable to compression than the FFN layers (Lv et al., 2024). We show PAMM is capable of reducing the memory consumption of these activations by a factor of $\times 512$, effectively eliminating their memory footprint during training. Importantly, PAMM leaves the forward pass and the gradients of other layers untouched, which accounts for its negligible impact on model performance..

PAMM is easily composable with most existing efficient training techniques, such as gradient checkpointing (Chen et al., 2016), low-rank adapters (Hu et al., 2021), and most importantly, efficient scaled-dot-product techniques (Choromanski et al., 2022; Dao, 2023). This makes PAMM a practical, easy-to-use plug-in for any model that uses scaled dot product attention.

In Section 3, we introduce PAMM as a general matmul approximation technique and present its complexity analysis and theoretical properties. In Section 4, we provide empirical evidence that PAMM drastically reduces the memory footprint of the $Q, K, V$ projections while maintaining model performance, in both pretraining (4.2) and finetuning (4.3). We provide a thorough throughput analysis in Section 4.4, and demonstrate that other compression methods don't scale nearly as well as PAMM in Section 4.6.

## 2    RELATED WORK

**Memory Efficient Attention**    Many recent works identify redundancies in the attention mechanism and exploit them to save time and memory during training. GQA (Ainslie et al., 2023) is commonly used in leading LLMs such as Mistral-7B (Jiang et al., 2023) and LLaMA (et al., 2024a) to reduce time and memory consumption by sharing the key and value tensors across multiple queries. Linformer (Wang et al., 2020) achieves linear time and memory by applying learned linear projections to the sequence dimension of the key and value matrices. Reformer (Kitaev et al., 2020) leverages hashing to approximate attention in $O(n \log n)$ time. Performer (Choromanski et al., 2022) replaces softmax attention with positive random feature maps to kernelize the attention and achieve linear complexity. Longformer (Beltagy et al., 2020) combines these with a sliding-window local attention with a small set of global tokens, yielding $O(nw)$ complexity (where $w$ is window size) that remains efficient for very long sequences. All of these works approximate the scaled-dot-product mechanism itself, and are therefore complementary to our work, which reduces the memory consumption of the linear layers that precede attention. In particular, (Choromanski et al., 2022) applies low-rank projections to the hidden embedding dimension, not the sequence dimension, which could be harmful to model performance as it exploits a less redundant axis.

**Low Rank Attention at Inference time**    Several works propose replacing the linear projections with low-rank approximations to reduce inference cost, including DeepSeek-V2 (et al., 2024b) which introduced Multi-head Latent Attention (MLA), Deepseek's Native Sparse Attention (NSA) (Yuan et al., 2025) and SinkLora (Zhang, 2024); these methods primarily target compression of the $KV$ cache at inference. In this work, we focus on pretraining and finetuning, making these works mostly irrelevant to our discussion. ESPACE (Sakr & Khailany, 2024) also employs a matrix-multiplication approximation, but with a fundamentally different objective: it introduces static projection matrices during training to enable post-training weight compression. This approach does not reduce training memory and may even increase it, since the projection matrices must be stored. In contrast, PAMM directly reduces attention memory during training without modifying model weights or inference-time behavior.

**Memory Efficient Training**    Numerous works aim at improving the efficiency of LLM training without focusing on the attention mechanism, such as quantization methods (Pan et al., 2022; Liu et al., 2022; Dettmers et al., 2023; Xi et al., 2024; Zhang et al., 2024), parallelism techniques (Huang et al., 2019; Shoeybi et al., 2020; Li et al., 2014; Dean et al., 2012), mixed precision training (Micikevicius et al., 2017), and low rank approximation methods (Shamshoum et al., 2025; Zhao et al., 2024; Hu et al., 2021; Lialin et al., 2023; Han et al., 2024). These approaches are either orthogonal to PAMM or can be applied in conjunction with it.

**Approximate Backward Pass**    Various methods have been proposed to reduce memory consumption of LLM training by approximating the backward pass. (Adelman et al., 2021; Liu et al., 2024) perform an approximate activation-gradient multiplication during backpropagation by randomly sampling rows and columns of the tensors. Approx-BP (Yang et al., 2024) reduces memory during backpropagation by approximating nonlinearities and introducing memory-sharing normalization layers. CompAct (Shamshoum et al., 2025) compresses the computation graph - activations, gradients, and optimizer states - using Gaussian random projections. Unlike CompAct, PAMM performs a simple data-dependent compression that operates along the batch and sequence dimensions. This difference is key: by exploiting redundancy across the batch rather than the embedding dimension, PAMM achieves substantially higher compression rates without degrading perplexity. We compare variations of these methods in Section 4.6.

## 3    PAMM: POINT-APPROXIMATE MATRIX MULTIPLICATION

### 3.1    METHOD OVERVIEW

Consider a linear layer $Z = X \cdot W$, e.g. the $Q$ projection of one layer of attention in an LLM. Here $X \in \mathbb{R}^{BL \times n}, W \in \mathbb{R}^{n \times m}, Z \in \mathbb{R}^{BL \times m}$, where $B$ is the batch size and $L$ is the sequence length during training. For brevity, we denote the total number of tokens in a batch as $b = BL$. During backpropagation, we receive the gradient of the loss w.r.t $Z$, denoted $\nabla Z$. Backpropagation

for this layer produces two new gradients: $\nabla X = \nabla Z \cdot W^\top$ and $\nabla W = X^\top \cdot \nabla Z$. Thus, the input activations $X$ must be stored during the forward pass for use in the backward computation. To mitigate this memory bottleneck, we introduce a novel tensor compression technique, *Point-Approximate Matrix Multiplication* (PAMM). Instead of storing the full input $X$, PAMM stores a much smaller compressed representation and computes an approximate gradient $\widetilde{\nabla W}$ during the backward pass as shown in Figure 2.

PAMM leverages the fact that the number of tokens in a training batch, $b$, is typically much larger than the hidden dimension $n$. For example, when pretraining LLaMA-1B, $b = 16384$ and $n = 2048$. Consequently, $\text{rank}(X) \leq n$, meaning that the rows of $X$ lie in a much lower-dimensional subspace. In principle, one could represent $X$ using only $n$ basis rows - an 8× reduction in this example - without any information loss. Accurately computing this aggressive reduction is costly, requiring a full Singular Value Decomposition, $\text{SVD}(X) = U\Sigma V^\top$, which is both time-consuming and memory intensive, as $U \in \mathbb{R}^{b \times n}$ is the same size as $X$.

PAMM provides an efficient alternative. We treat rows of $X$ as points in $\mathbb{R}^n$ and compress $X$ by finding closely matching points in $\mathbb{R}^n$ that are generated from a small subset of the original rows of $X$, used as the generating set, which we denote $C$. This allows us to only store the subset $C$ and two auxiliary vectors required to generate the approximated $X$ from $C$. The size of $C$ is $k = r \cdot b$, where $r$ is the compression ratio. Empirically, we find that extremely small values of $r$ (as low as $1/512$) maintain model performance while reducing the memory used by the input activation tensor to nearly zero.

We present PAMM for general matrix multiplication in Section 3.2.

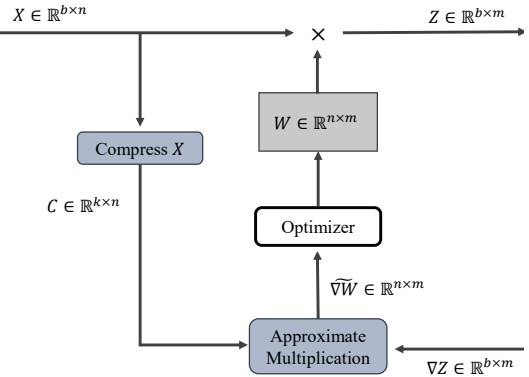

Figure 2: **Overview of training with PAMM.** In a standard linear layer, backpropagation multiplies the upstream gradient by the input $X$ to compute the gradient with respect to the parameters $\nabla W = X^\top \cdot \nabla Z$. In our method, for the $W_Q, W_K, W_V$ projections, we store a compressed version of $X$ and approximate the gradient-input matrix multiplication with PAMM. Here, $b$ denotes the total number of tokens in a training batch. Pseudocode is provided in Appendix A.

## 3.2 Algorithm Description

Consider two matrices $A \in \mathbb{R}^{b \times n}$ and $B \in \mathbb{R}^{b \times m}$, their product $O = A^\top B$, and let $A_i \in \mathbb{R}^n$ denote the $i$-th row of $A$. PAMM efficiently approximates $O$ in two stages: (1) compressing $A$ into a small set of *generators* and per-row coefficients, and (2) using this compressed representation to compute an efficient approximation $\tilde{O}$. This is illustrated in Figure 1 and a full pseudocode is provided in Appendix A.

**Compressing $A$:** Let $\{C_j\}_{j=1}^k$ denote a set of $k$ *generating points*, or *generators* in $\mathbb{R}^n$, which are the rows of a matrix $C \in \mathbb{R}^{k \times n}$. These can be any points "close enough" to the points in $A$. In our implementation, we obtain $C$ by sampling $k$ rows from $A$ without replacement.

Given a point $A_i$, we choose its representative as the *closest point on the line spanned by any of the generators in $C$*. For each generator $C_j$ and point $A_i$, we denote the closest approximation point to

$A_i$ on the line spanned by $C_j$ as

$$\tilde{A}_{i,j} = \frac{\langle A_i, C_j \rangle}{\|C_j\|_2^2} \cdot C_j = \alpha(i,j)C_j. \tag{1}$$

Next, we choose the closest approximation point $\tilde{A}_{i,j}$ to $A_i$ as its representative, with $f(i)$ an assignment function: $f(i) = \arg\min_j \|A_i - \tilde{A}_{i,j}\|_2$. That is, for each point $A_i$ we choose the representative $\tilde{A}_i = \tilde{A}_{i,f(i)} = \alpha(i, f(i)) \cdot C_{f(i)}$. The assignment function $f$ selects which generator $C_j$ is used to generate $\tilde{A}_i$.

Note that $\alpha(i,j) = \text{csim}(A_i, C_j) \cdot \frac{\|A_i\|_2}{\|C_j\|_2}$, where $\text{csim}(x,y)$ is the cosine similarity between $x$ and $y$. This points us to the following lemma, which we prove in Appendix B:

**Lemma 1.** *The generator representing $A_i$ is the one with the highest absolute cosine similarity to $A_i$, i.e. $f(i) = \arg\max_j |\text{csim}(A_i, C_j)|$.*

Lemma 1 allows us to efficiently compute the assignment function $f(i)$: we start by computing the cosine similarity matrix, $\text{csim}(A, C) \in \mathbb{R}^{b \times k}$, which involves one matmul $AC^\top$ and normalization over the rows. We then compute an $\arg\max$ operation over the columns of $\text{csim}(A, C) \in \mathbb{R}^{b \times k}$ to obtain $f(i)$.

Finally, to control the approximation quality, we introduce a tolerance value $\varepsilon \geq 0$ and require:

$$\|A_i - \tilde{A}_i\|_2 \leq \varepsilon \|A_i\|_2. \tag{2}$$

We call this condition the *neighborhood condition*, as it requires the chosen representatives to lie in an $\varepsilon$-neighborhood around the points $A_i$. If the best representative we could find for $A_i$ doesn't fulfill this requirement, we simply drop $A_i$ in the final approximation. This is equivalent to choosing the zero vector $\mathbf{0} \in \mathbb{R}^n$ as $A_i$'s representative or setting $\alpha_i = 0$.

This allows us to replace the entire matrix $A$ with a set of generating points $\{C_j\}_{j=1}^k$, a list of indices $f = \{f(i)\}_{i=1}^b$ and a list of coefficients $\alpha = \{\alpha_i\}_{i=1}^b$ where $\alpha_i = \alpha(i, f(i))$.

**Approximate Matrix Multiplication:** We wish to approximate $O = A^\top B$. Given the compressed results of the previous stage $C$, $f$ and $\alpha$, the approximate matrix $\tilde{A} \in \mathbb{R}^{b \times n}$ can be reconstructed as follows:

$$\tilde{A}_i = \begin{cases} \alpha_i \cdot C_{f(i)} & A_i \text{ satisfies Eq. 2,} \\ 0 & \text{otherwise.} \end{cases} \tag{3}$$

Reconstructing $\tilde{A}$ would allow us to compute the direct estimate of the original product $\tilde{O} = \tilde{A}^\top B$. However, this product would require as much compute as the original $O$. Instead, we exploit the structure induced by the generators for efficiency. The approximate matrix multiplication can be written as a sum of the product of rank-one matrices $\tilde{A}_i^\top \in \mathbb{R}^{n \times 1}$ and $B_i \in \mathbb{R}^{1 \times m}$, where each $\tilde{A}_i$ can be expanded using the representatives in Eq. 3:

$$\tilde{O} = \tilde{A}^\top B = \sum_{i=1}^b \tilde{A}_i^\top B_i = \sum_{i=1}^b \alpha_i C_{f(i)}^\top B_i.$$

Since each generator $C_j$ approximates several points, we can rewrite this sum as

$$\tilde{O} = \sum_{j=1}^k C_j^\top \cdot \left( \sum_{i:f(i)=j} \alpha_i B_i \right) = \sum_{j=1}^k C_j^\top \cdot \tilde{B}_j.$$

As a result, instead of reconstructing $\tilde{A} \in \mathbb{R}^{b \times n}$, we can compute the matrix $\tilde{B} \in \mathbb{R}^{k \times m}$, and compute $\tilde{O} = C^\top \tilde{B}$, which is a much cheaper multiplication than the original $O = A^\top B$.

**Normalization factor $\beta$:** Since some rows were dropped $\alpha_i = 0$, $\tilde{O}$ underestimates the full sum in expectation. To mitigate this, we introduce a correction factor $\beta$ into the computation of $\tilde{O}$ such that $\mathbb{E}[\tilde{O}] = O$. Assuming the rows in $A$ are drawn from some distribution $A_i \sim \mathbb{P}_A$ over $\mathbb{R}^n$, if we assume that a row $A_i$ is kept $\iff$ its representative is close enough, e.g. $\tilde{A}_i \sim A_i$, then $\tilde{O}$ can be written as:

$$\tilde{O} \approx \beta \sum_{i=1}^{b} M_i A_i^\top B_i, \qquad M_i = \begin{cases} 1, & \text{row } i \text{ is kept,} \\ 0, & \text{otherwise.} \end{cases} \tag{4}$$

If $\eta$ is the number of dropped rows, we have $\Pr[M_i = 1] = \frac{b-\eta}{b}$. Hence in expectation:

$$\mathbb{E}[\tilde{O}] \approx \beta \sum_{i=1}^{b} \mathbb{E}[M_i] A_i^\top B_i = \beta \frac{b-\eta}{b} \sum_{i=1}^{b} A_i^\top B_i = \beta \frac{b-\eta}{b} O. \tag{5}$$

Choosing $\beta = \frac{b}{b-\eta}$ ensures $\mathbb{E}[\tilde{O}] = O$. This $\beta$ value is stored along with the rest of the compression outputs. Note that requiring $\tilde{A}_i \sim A_i$ implies using a small value for $\varepsilon$, which means fewer points will satisfy the neighborhood condition, increasing $\eta$.

### 3.2.1 Theoretical Guarantees of PAMM

Let $\mathcal{I}_j = \{i \in [b] | f(i) = j\}$ be the index set of all the rows $A_i$ whose representative $\tilde{A}_i$ is generated by the point $C_j$. The union of these sets, $\mathcal{I} = \cup_{j \in [k]} \mathcal{I}_j$, is the set of all indices for which $\tilde{A}_i$ satisfies the neighborhood condition, i.e. the indices of points that were not dropped. Denote these points as $A_\mathcal{I}$, and all the dropped rows as $A_{\overline{\mathcal{I}}}$. By construction of $\tilde{A}$ we have

$$\|A - \tilde{A}\|_F^2 = \sum_{j \in [1,k]} \sum_{i \in \mathcal{I}_j} \|A_i - \tilde{A}_i\|^2 + \sum_{i \notin \mathcal{I}} \|A_i\|^2$$
$$\leq \varepsilon^2 \sum_{j \in [1,k]} \sum_{i \in \mathcal{I}_j} \|A_i\|^2 + \sum_{i \notin \mathcal{I}} \|A_i\|^2 = \varepsilon^2 \|A_\mathcal{I}\|_F^2 + \|A_{\overline{\mathcal{I}}}\|_F^2.$$

As for the approximation error, due to submultiplicativity we get

$$\|O - \tilde{O}\|_F^2 = \|(A - \tilde{A})^\top B\|_F^2 \leq \|B\|_2^2 (\varepsilon^2 \|A_\mathcal{I}\|_F^2 + \|A_{\overline{\mathcal{I}}}\|_F^2).$$

This implies that the points without representatives could drastically hurt the approximation error. We analyze empirically the effect of the neighborhood condition on approximation error in Appendix H.

Next, we present a lemma for choosing the appropriate number of generators $k$ to ensure all points in $A$ are represented:

**Lemma 2** ($k$ bound under uniform sampling). *Let $A \in \mathbb{R}^{b \times n}$ and $B \in \mathbb{R}^{b \times m}$. Fix a tolerance $0 < \varepsilon < 1$ and a failure probability $0 < \delta < 1$. Suppose we sample $k$ generators without replacement following a uniform distribution on the rows of $A$. The following bound on $k$ is sufficient for the generating set $C$ to fully cover $A$:*

$$k > \frac{b}{n_{\min}} \ln\left(\frac{b}{\delta}\right) \Rightarrow \Pr[\mathcal{I} = [b]] > 1 - \delta$$

Here $n_{min}$ is the size of the smallest neighborhood of points close enough to any row $A_i$, which represents the density of the data.

The lemma shows that as the total number of tokens in the batch $b = BL$ increases, we only need logarithmically more generators to capture the data distribution well. $n_{min}$ can be shown to be proportional to $b$, i.e. $b/n_{min}$ is approximately constant in $b$, which means that $k > c_0 \ln\left(\frac{b}{\delta}\right)$ for some constant $c_0$. A proof of the Lemma 2 can be found in Appendix C, including a proof that $b/n_{min}$ is approximately constant in $b$.

This observation accounts for PAMM's strong performance even when $k \ll b$. In our experiments (Section 4), compressing the activation by a factor as small as $1/512$ yields no loss in model performance and, in several cases, slightly improves the results when compared to training without PAMM.

## 4 EXPERIMENTS

In this section, we evaluate PAMM on model pretraining (Section 4.2) and finetuning (Section 4.3). Throughout, PAMM is applied to the $Q$, $K$, and $V$ projections of *every* attention block in the trained models. We demonstrate that even with an extremely small compression ratio - down to $1/512$ - PAMM preserves or can even improve performance across model sizes. In Section 4.4 we provide a thorough evaluation of PAMM's effect on training throughput and runtime. In Section 4.6, we compare with different compression algorithms, such as CompAct Shamshoum et al. (2025) and Uniform-CRS - a sampling technique similar to (Adelman et al., 2021; Liu et al., 2024). We demonstrate that these methods can't achieve the same memory reduction without degrading model performance significantly. All of the experiments were conducted on NVIDIA A100 GPUs.

### 4.1 CHOOSING $k$ AND $\varepsilon$ IN PRACTICE

As discussed in Section 3.1, PAMM leverages rank redundancy in the input tensor $X$ used by the $Q$, $K$, and $V$ projections. We parameterize the compression by a ratio $r \in (0, 1]$: given $b = BL$ rows - for mini-batch size $B$ and sequence length $L$ - we save $k = \lceil r \cdot b \rceil$ generators. In pre-training, we push $r$ as low as $1/512$; in some fine-tuning settings we even reach $k = 1$, i.e. a single generator $C_1$. Also, we experimented with a wide range of $\varepsilon$ values, and found that setting $\varepsilon \to \infty$, i.e., removing the neighborhood constraint altogether, gives the best performance. Another possible choice of $\varepsilon$ is $\varepsilon = 0$, which reduces PAMM to a Column-Row-Sampling algorithm with uniform choice of column-row pairs, which we call Uniform-CRS. In Section 4.6 we delve deeper into these comparisons and the effect of $\varepsilon$. We provide more visualizations of PAMM in Appendix H.

### 4.2 PRETRAINING

We evaluate PAMM in pretraining by applying it to the $Q, K, V$ projections of LLaMA models (Touvron et al., 2023), from LLaMA-60M up to LLaMA-7B. All models were pretrained on the C4 dataset (Raffel et al., 2023) without data repetition. We provide full results for LLaMA-7B in Appendix E, proving its effectiveness in larger models.

Our experiment setup follows those of Shamshoum et al. (2025); Zhao et al. (2024), where the learning rate is the only hyperparameter that is tuned for each model size. For evaluation we report model perplexity at the end of training. For more details on the training experiments, we refer to Appendix D, and for stability analysis to Appendix I.

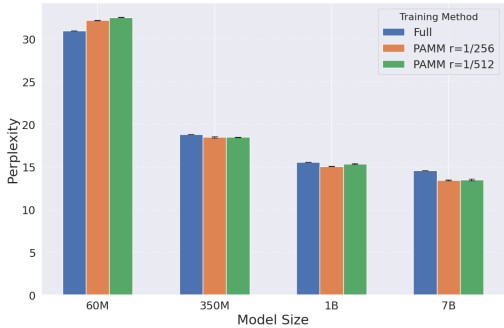 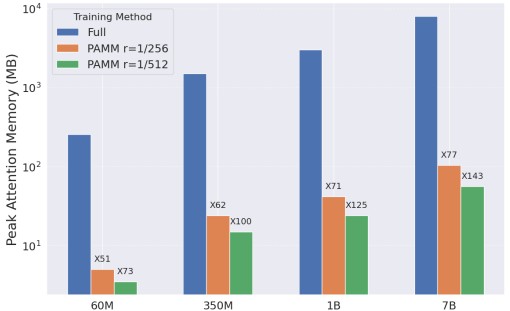

(a) **Pretraining perplexity across model sizes.** We report validation perplexity for LLaMA-60M-7B, trained with PAMM at different compression ratios, compared to the full-rank baseline. Error bars in black show minimal variance.

(b) **Peak attention memory across model sizes.** Memory corresponds to activations of all $Q, K, V$ projection layers. PAMM reduces memory usage by over 97% at all model sizes while preserving perplexity close to the full baseline.

Figure 3: **Model perplexity and attention memory when pretraining LLaMA models on C4 with PAMM.** PAMM achieves massive memory reductions while maintaining or even increasing perplexity compared to the baseline. Results averaged over k=3 runs per configuration.

Figure 3 shows that applying PAMM to the $Q$, $K$, and $V$ projections compresses the activations by a factor of $512\times$ while leaving model quality essentially unchanged. For larger models, perplexity ac-

tually *decreases* relative to the full-memory baseline, suggesting that redundant rows in the attention inputs can hinder training. This observation supports the redundancy we anticipated, as discussed in Section 3.1. Exact measurement results are reported as a table in Appendix D of the supplementary materials.

## 4.3 FINETUNING

For finetuning, we apply PAMM to the $Q, K, V$ projections of RoBERTa-base (Liu et al., 2019) and evaluate on the GLUE benchmark (Wang et al., 2019). We follow the finetuning protocol of Shamshoum et al. (2025); Zhao et al. (2024) and tune only the learning rate per task.

For all tasks we report performance by taking the average of 3 runs. Performance is measured as F1 score for MRPC, Pearson's Correlation for STS-B, Matthew's correlation for CoLA, and classification accuracy for all other tasks. We measure the peak GPU memory consumption of all $Q, K, V$ projections in the model in MB. For more details, we refer to Appendix G.

Table 1: **Finetuning RoBERTa-base on the GLUE benchmark with PAMM.** We compare full fine tuning and PAMM with two different compression rates. Reported memory in MB corresponds to the memory consumed by the activations of the $Q, K, V$ projection layers. PAMM preserves the model's performance while reducing memory consumption by more than 97%.

|  | Memory (MB) | CoLA | STS-B | MRPC | RTE | SST2 | MNLI | QNLI | QQP | Avg |
|---|---|---|---|---|---|---|---|---|---|---|
| **Full Fine-Tuning** | **288** | 62.24 | 90.92 | 91.30 | 79.42 | 94.57 | 87.18 | 92.33 | 92.28 | **86.28** |
| PAMM $r = 1/128$ | **6.75** | 63.17 | 90.90 | 92.12 | 78.5 | 93.84 | 86.64 | 92.17 | 91.579 | **86.11** |
| PAMM $r = 1/256$ | **3.37** | 62.49 | 90.75 | 92.34 | 78.4 | 94.22 | 87.08 | 92.59 | 91.575 | **86.18** |

As shown in Table 1, PAMM drastically reduces the memory consumption of the $Q, K, V$ projections, by 2 orders of magnitude, while achieving competitive performance across tasks and on average.

## 4.4 THROUGHPUT

In this section we provide empirical measurements for PAMM's effect on runtime and overall training throughput. Overall, PAMM does not significantly reduce throughput, and the overhead is negligible in the context of full model training, especially for larger models.

We measure throughput as the number of input tokens processed per second, per GPU, per full training iteration. We use a batch size of 128K tokens averaged over 1000 training iterations, measuring throughput for PAMM with $r = 1/512$ and baseline, on models of different sizes. For the models LLaMA-1B and LLaMA-7B we used DDP training with 8 GPUs, with the same global batch size of 128K tokens, meaning each GPU sees 16K tokens. Results are shown in Table 2a. The results show that PAMM's throughput degradation decreases significantly as the model size increases and remains below 2.7% for the 1B model and 2.1% for the 7B model.

Table 2: **Throughput and runtime comparisons of models with and without PAMM.**

(a) **Throughput across model sizes.** PAMM introduces minimal degradation, remaining below 2.7% for the 1B model and 2.1% for the 7B model.

| Model Size | PAMM (tok/sec) | Baseline (tok/sec) | Throughput Degradation |
|---|---|---|---|
| 60M | 122k | 152k | **19.74%** |
| 350M | 181k | 205k | **11.71%** |
| 1B | 73k | 75k | **2.67%** |
| 7B | 20.05K | 20.48K | **2.1%** |

(b) **Throughput comparison in tok/sec of forward (FP) and backward (BP) passes of LLaMA-1B.** The final column reports runtime degradation relative to baseline. The total runtime degradation is very small.

|  | Baseline | PAMM Total | Throughput Degradation |
|---|---|---|---|
| Forward | 247.6K | 235.4K | 4.92% |
| Backward | 141.9K | 138.3K | 2.53% |
| Total | 88.4K | 85.2K | 3.61% |

For LLaMA-1B we also report a detailed runtime breakdown of the specific operations in PAMM in Appendix F.

We can see that our current implementation introduces only a small runtime overhead (5.1% in FP, and 3.5% in BP). The additional compute required for PAMM is modest relative to the overall cost of a training step. In short, this overhead is negligible in the context of full model training and quickly decreases with model size. An analysis of PAMM's time complexity is provided in Appendix J.

## 4.5 ABLATION STUDY - BATCH SIZE AND SEQUENCE LENGTH

We ran several experiments with different batch sizes and sequence lengths and measured model perplexity with and without PAMM. We used the LLaMA-60M model since we already demonstrated PAMM scales well to larger models. In all experiments, we used $r = 1/512$. Results are shown in Table 3. The results show that PAMM's perplexity is comparable with the baseline in all configurations, demonstrating that PAMM performs well regardless of the choice of batch size or sequence length.

Table 3: **Training perplexity of Baseline vs. PAMM across batch sizes and sequence lengths.**

| Batch Size | Sequence Length | Baseline Perplexity | PAMM Perplexity | Relative change |
|------------|-----------------|---------------------|-----------------|-----------------|
| 128 | 256 | 42.63 | 43.01 | +0.8% |
| 128 | 1024 | 37.47 | 37.03 | -1.1% |
| 256 | 256 | 37.61 | 37.25 | -0.9% |
| 256 | 512 | 33.32 | 33.73 | +1.2% |
| 512 | 128 | 37.38 | 36.43 | -2.5% |
| 512 | 256 | 30.97 | 32.46 | +4.8% |
| 512 | 512 | 29.16 | 30.30 | +3.9% |

## 4.6 ABLATION STUDY - RANK AND $\varepsilon$

In this section we answer two questions, using a LLaMA-60M model: (1) How does PAMM compare to other existing techniques that compress the activations of the $Q, K, V$ projections? (2) How does varying $\varepsilon$ during training affect performance?

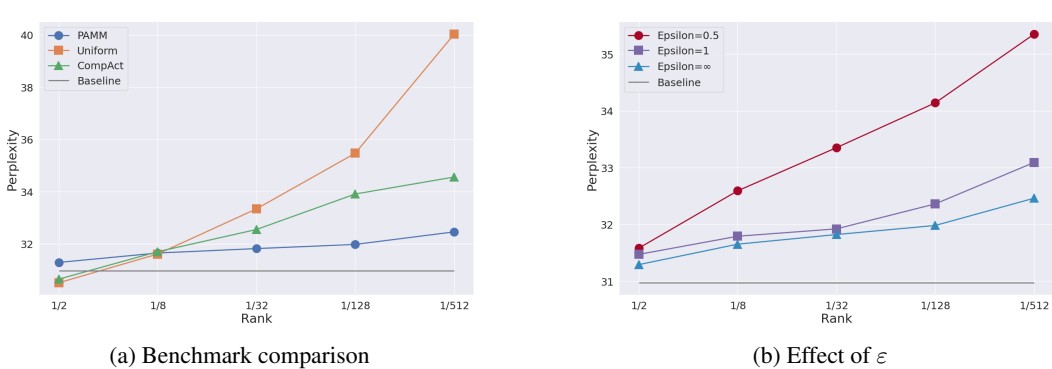

(a) Benchmark comparison        (b) Effect of $\varepsilon$

Figure 4: (a) **Compression technique comparison** when applied to $Q, K, V$ projections of LLaMA-60M, with different compression rates. PAMM significantly outperforms other compression methods and can utilize a very low $r$. (b) **Effect of $\varepsilon$ on performance of LLaMA-60M with PAMM.** When $\varepsilon = 0$, PAMM is equivalent to Uniform-CRS; when $\varepsilon = \infty$, PAMM doesn't enforce the neighborhood condition. We can see that choosing $\varepsilon = \infty$ is the best option for all compression rates checked.

We benchmark PAMM against *CompAct* (Shamshoum et al., 2025) and *Uniform-CRS*, a simple subsampling technique similar to (Adelman et al., 2021), which is equivalent to PAMM with $\varepsilon = 0$ as mentioned in Section 4.1. CompAct stores a low-rank sketch $\tilde{X} = XP$ where $P \in \mathbb{R}^{n \times k}$ is random, $k = rb$, and $\mathbb{E}[P^\top P] = I$; gradients are unbiased but noisy. Uniform-CRS is the simplest form of PAMM: keeps only the sampled generators, discarding all others. Figure 4a shows perplexity versus memory as $r$ shrinks. PAMM delivers up to $512\times$ memory savings with negligible

perplexity change, whereas CompAct and Uniform-CRS suffer pronounced degradation, underscoring PAMM's superior memory-quality trade-off.

Finally, in order to choose the value of $\varepsilon$, we measured model perplexity for different choices of $\varepsilon$. Results are reported in figure 4b. The best accuracy is obtained at $\varepsilon = \infty$, i.e. when every row is represented by a generator; smaller $\varepsilon$ values exclude more rows and increase perplexity. The small gap between $\varepsilon = 1$ and $\infty$ indicates that attention inputs are already relatively clustered, which explains why dropping too many points harms performance.

### 4.7 MULTI-MODAL AND PEFT COMPATIBILITY

We also finetune a Pixtral-12B Vision-Language Model (VLM) (Agrawal et al., 2024) on the AID satellite image classification benchmark (Xia et al., 2017), with and without PAMM. This experiment demonstrates: (1) PAMM's scalability to even larger models, (2) PAMM's compatibility with PEFT methods (LoRA), and (3) PAMM's applicability to multi-modal models. As we show in Table 4, PAMM maintains performance while reducing non-negligible memory.

The model is given an image and a textual instruction prompt asking it to produce a final classification, as standard in Visual Question Answering (VQA). Performance is evaluated using F1 averaged over the 30 possible classes. For the baseline, we finetune the model using LoRA with rank 128, applied to the language model and vision encoder, where $\text{LoRA}(x) = (W_0 + AB)(x)$, $W_0$ is the pretrained weight and $A, B$ are learned low rank matrices. PAMM is combined with LoRA by compressing the input activation of the $Q, K, V$ as always, which is equivalent to applying PAMM to the $A$ layer of the LoRA adapter. We note PAMM can also be applied to the $B$ linear layer, but that would result in marginal memory gains as $B$ stores a much smaller input activation.

| Model | Macro F1 | Weighted F1 | $Q, K, V$ Mem. Saved |
|---|---|---|---|
| Pretrained Baseline | 0.51 | 0.54 | — |
| +LoRA FT | $0.9711 \pm 0.0028$ | $0.9724 \pm 0.0030$ | $0\%$ |
| +LoRA+PAMM $r = 1/128$ | $0.9727 \pm 0.0027$ | $0.9737 \pm 0.0022$ | $97.65\%$ |
| +LoRA+PAMM $r = 1/512$ | $0.9691 \pm 0.0015$ | $0.9705 \pm 0.0015$ | $99.28\%$ |

Table 4: **Pixtral-12B VLM results on the AID satellite image classification benchmark with and without PAMM.** We report Macro F1 (F1 averaged across classes) and Weighted F1 (class-frequency-weighted F1 average). PAMM at rank $r = 1/128$ uses $k = 2$ in the vision encoder and $k = 4$ in the language model, while $r = 1/512$ uses $k = 1$ for both. We also report the memory saved (in MB and as a percentage of the $Q, K, V$ activation memory required in standard LoRA finetuning). Each configuration is evaluated over 3 runs to compute standard deviations. PAMM effectively removes the $Q, K, V$ activation memory cost during finetuning while maintaining competitive performance.

## 5 CONCLUSION

We presented PAMM, a memory efficient technique, and demonstrated that it drastically reduces the memory footprint of the input activations to the $Q, K, V$ projections in attention layers. PAMM is easy to implement, simple to use and easily composable with other efficient training techniques. Future work could attempt to extend PAMM to approximate the activations of other layers in the transformer model, or to improve tradeoffs by choosing the generating set $C$ more strategically.

## REPRODUCIBILITY STATEMENT

The full source code for PAMM is publicly available at `https://gitlab.com/MalikKhalaf4/pamm`. In the appendices, we include full pseudocode, detailed experimental settings and results, and complete theoretical proofs.

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

# A PAMM ALGORITHM

In this appendix, we provide detailed pseudo-code for PAMM. Algorithm 1 describes PAMM as a two stage approximate matrix multiplication method given two input matrices $A \in \mathbb{R}^{b \times n}, B \in \mathbb{R}^{b \times m}$; first, compressing $A$, and then approximating the product $\tilde{O}$. Algorithms 2 and 3 describe the forward and backward pass of a linear layer with PAMM.

---

**Algorithm 1** PAMM: Point-Approximate Matrix Multiplication

---

1: **procedure** COMPRESS($A \in \mathbb{R}^{b \times n}, \ k, \ \varepsilon$)
2: **Input:** $A \in \mathbb{R}^{b \times n}$, number of generators $k$, tolerance $\varepsilon \geq 0$.
3: **Output:** Compressed representation of $A$
4:      Sample indices $\mathcal{I} \subset [b]$ uniformly without replacement, $|\mathcal{I}| = k$.
5:      $C \leftarrow A[\mathcal{I}, :]$                       $\triangleright$ generators, $C \in \mathbb{R}^{k \times n}$
6:      Initialize $\alpha \leftarrow \mathbf{0}^b, \ f \leftarrow \mathbf{0}^b$
7:      $\eta_A \leftarrow \|A\|_{\text{rows}} = \text{norm}(A, \dim=1) \in \mathbb{R}^{b \times 1}$        $\triangleright$ Array of row-norms of $A$
8:      $\eta_C \leftarrow \eta_A[\mathcal{I}] \in \mathbb{R}^{k \times 1}$        $\triangleright$ Array of row-norms of $C$ (slice from array $\eta_A$)
9:      $\text{csim}(A, C) \leftarrow \frac{A \cdot C^\top}{\eta_A \cdot \eta_C^\top} \in \mathbb{R}^{b \times k}$        $\triangleright$ Division is element-wise
10:      $f \leftarrow \arg\max \left( \text{csim}(A, C), \dim=1 \right) \in [k]^b$
11:      $\alpha \leftarrow \text{csim}(A, C)[:, f] \odot \frac{\eta_A}{\eta_C[f]} \in \mathbb{R}^b$
12:      $e = \|A - \alpha \odot C[f, :]\|_2 \in \mathbb{R}^b$        $\triangleright$ Broadcast $\alpha$ along the column dimension $n$
13:      $\alpha[e > \varepsilon] \leftarrow 0$        $\triangleright$ Neighborhood Condition via masked assignment
14:      **return** $C, \alpha, f$
15: **end procedure**

---

1: **procedure** APPROXMM
2: **Input:** $C, \alpha, f$, and $B \in \mathbb{R}^{b \times m}$
3: **Output:** Approximate multiplication $\tilde{O} \in \mathbb{R}^{n \times m}$
4:      Initialize $\widetilde{B} \leftarrow \mathbf{0} \in \mathbb{R}^{k \times m}$
5:      $B' \leftarrow \alpha \odot B$        $\triangleright$ Multiply each row of $B$ with its $\alpha$ value
6:      $\widetilde{B} \leftarrow \text{index\_add}(\widetilde{B}, 0, f, B')$        $\triangleright$ Efficient parallel computation of $\sum_{i: f(i)=j} \alpha_i B_{i,:}$
7:      $\widetilde{O} \leftarrow C^\top \widetilde{B}$        $\triangleright$ $\widetilde{O} \in \mathbb{R}^{n \times m}$
8:      **return** $\widetilde{O}$
9: **end procedure**

---

**Algorithm 2** Linear Layer: Forward Pass with PAMM

---

**Input:** An input $X \in \mathbb{R}^{B \times L \times n}$, a weight $W \in \mathbb{R}^{n \times m}$, and a rank $r$.
1: $Z \leftarrow XW \in \mathbb{R}^{B \times L \times m}$
2: $C, \alpha, f \leftarrow \text{Compress}(\text{flatten}(X), r)$
3: **save\_for\_backward**$(C, \alpha, f, W)$
4: **return** $Z$

---

**Algorithm 3** Linear Layer: Backward Pass with PAMM

---

**Input:** A gradient $\nabla Z \in \mathbb{R}^{B \times L \times m}$, A compressed representation of the activation $C, \alpha, f$ and a weight $W \in \mathbb{R}^{n \times m}$.
1:
2: $\widetilde{\nabla W} \leftarrow \text{ApproxMM}(C, \alpha, f, \text{flatten}(\nabla Z)) \in \mathbb{R}^{n \times m}$
3: $\nabla X \leftarrow \nabla Z \cdot W^\top \in \mathbb{R}^{B \times L \times n}$
4: **return** $\nabla X, \widetilde{\nabla W}$

---

# B    PROOF OF LEMMA 1

*Proof.*

$$\arg\min_{j} \|A_i - \tilde{A}_{i,j}\|_2 \tag{6}$$

$$= \arg\min_{j} \|A_i - \mathrm{csim}(A_i, C_j) \cdot \frac{\|A_i\|_2}{\|C_j\|_2} C_j\|_2 \tag{7}$$

$$= \arg\min_{j} \left\{ \|A_i\|_2 \cdot \| \frac{A_i}{\|A_i\|_2} - \mathrm{csim}(A_i, C_j) \cdot \frac{C_j}{\|C_j\|_2} \|_2 \right\} \tag{8}$$

$$= \arg\min_{j} \left\{ \|A_i\|_2 \left( \underbrace{\frac{\langle A_i, A_i \rangle}{\|A_i\|_2^2}}_{=1} - 2\mathrm{csim}(A_i, C_j) \cdot \underbrace{\frac{\langle A_i, C_j \rangle}{\|A_i\|_2 \|C_j\|_2}}_{=\mathrm{csim}(A_i, C_j)} + \mathrm{csim}(A_i, C_j)^2 \underbrace{\frac{\langle C_j, C_j \rangle}{\|C_j\|_2^2}}_{=1} \right) \right\} \tag{9}$$

$$= \arg\min_{j} \left\{ \|A_i\|_2 (1 - \mathrm{csim}(A_i, C_j)^2) \right\} \tag{10}$$

$$= \arg\max_{j} |\mathrm{csim}(A_i, C_j)| \tag{11}$$

$\square$

For the last step, since the $\min$ operation is over $j$, $\|A_i\|_2$ is constant and can be removed. Also, the $\min$ becomes $\max$ because of the sign change.

# C    PROOF OF LEMMA 2

First, for $x, y \in \mathbb{R}^n$, define $h(x, y) \in \mathbb{R}^n$ to be the point in $\mathrm{Span}\{y\}$ closest to $x$, i.e. $h(x, y) = \frac{\langle x, y \rangle}{\|y\|^2} \cdot y$. Note that in Section 3.2 we had $\tilde{A}_{i,j} = h(A_i, C_j)$. Also define $N_\varepsilon(i) = \{A_j \mid \|A_i - h(A_i, A_j)\|_2 \le \varepsilon \|A_i\|_2\}$ to be the $\varepsilon$-*neighborhood* of $A_i$. In other words, it's the set of all points in $A$ capable of generating a representative for $A_i$ that will satisfy the neighborhood condition. Denote $n_{min} = \min_i |N_\varepsilon(i)|$ the size of the smallest $\varepsilon$-neighborhood in $A$. Note that $n_{min} \ge 1$, as $A_i$ can always generate itself with $\alpha = 1$. We also remind the reader that $\mathcal{I}_j = \{i \in [b] \mid f(i) = j\}$ is the index set of all rows whose representative is generated by $C_j$, and that $\mathcal{I} = \cup_{j \in [k]} \mathcal{I}_j$ is the set of all indices which have representatives satisfying the neighborhood condition.

We now begin our proof.

*Proof.* Let $C$ be the set of generating points chosen uniformly at random from $A$, without replacement, and consider a row $A_i$. The probability that no $C_j$ generates a representative for $A_i$ which satisfies the neighborhood condition is the probability of selecting all $C_j$ outside of $N_\varepsilon(i)$:

$$\Pr[i \notin \mathcal{I}] = Pr[\forall C_j : C_j \notin N_\varepsilon(i)]$$

$$= \frac{\binom{b - |N_\varepsilon(i)|}{k}}{\binom{b}{k}} = \frac{\prod_{s=0}^{k-1} (b - |N_\varepsilon(i)| - s)}{\prod_{s=0}^{k-1} (b - s)}$$

$$= \prod_{s=0}^{k-1} \frac{b - |N_\varepsilon(i)| - s}{b - s} = \prod_{s=0}^{k-1} \left( 1 - \frac{|N_\varepsilon(i)|}{b - s} \right)$$

$$\le \prod_{s=0}^{k-1} \left( 1 - \frac{|N_\varepsilon(i)|}{b} \right) = \left( 1 - \frac{|N_\varepsilon(i)|}{b} \right)^k$$

Since $1 - x \le e^{-x}$, for all real $x$: $(1 - \frac{|N_\varepsilon(i)|}{b})^k \le \exp(-\frac{k|N_\varepsilon(i)|}{b})$. Thus:

$$\Pr[\exists i : i \notin \mathcal{I}] \le \sum_{i=1}^{b} \Pr[i \notin \mathcal{I}] \le b \exp\left(-\frac{k \cdot n_{\min}}{b}\right).$$

Demanding the above probability is sufficiently low:

$$b \cdot \exp\left(-\frac{k \cdot n_{min}}{b}\right) < \delta \quad \Rightarrow \quad k > \frac{b}{n_{min}} \ln\left(\frac{b}{\delta}\right)$$

Concluding our proof.

$\square$

**Proof of scaling of** $\frac{b}{n_{min}}$     In Section 3.2.1 we claim that $\frac{b}{n_{min}}$ is roughly constant with $b$, so $k$ scales like $\ln(\frac{b}{\delta})$. We will now prove this claim.

First we analyze the probability of some row $A_j$ to be inside the $\varepsilon$-neighborhood of another row $A_i$, i.e. $A_j \in N_\varepsilon(i)$. The neighborhood condition $\|A_i - \widetilde{A}_i\|_2 \le \varepsilon\|A_i\|_2$ defines an $L_2$-ball around $A_i$ with radius $\varepsilon\|A_i\|_2$: $\mathfrak{B}(A_i, \varepsilon\|A_i\|_2)$. Take the hypercone $V(A_i, \theta)$ rooted at the origin which is tangent to this ball. The angle spanning the hypercone is simply $\theta = 2\arcsin(\frac{\varepsilon\|A_i\|_2}{\|A_i\|_2}) = 2\arcsin(\varepsilon)$, which isn't dependent on the specific choice of $A_i$. In order for any point $x \in \mathbb{R}^n$ to also be in $N_\varepsilon(i)$, there must be some constant $c \in \mathbb{R}$ such that $cx$ is inside $\mathfrak{B}(A_i, \varepsilon\|A_i\|_2)$; therefore $x$ must lie within the hypercone $V(A_i, \theta)$, since it is the projective space of the ball $\mathfrak{B}(A_i, \varepsilon\|A_i\|_2)$.

Next, denote the data distribution of rows $A_j$ with $\mathcal{D}_A$. The probability of sampling a row $A_j$ inside $V(A_i, \theta)$ is $\Pr[A_j \in V(A_i, \theta)|A_j \sim \mathcal{D}_A] = z_i$. Assuming $A_j$ are i.i.d, when sampling $b$ points from $\mathcal{D}_A$ the expectation of the number of points inside $V(A_i, \theta)$ becomes simply $\mathbb{E}[|N_\varepsilon(i)|] = b \cdot z_i$.

Lastly, note that $z_i$ is a constant scalar value which doesn't depend on $b$; as $b$ grows, even though more rows need to be covered, the probability of finding another row close to the one we're considering stays the same. This proves the claim.

To put it another way, assuming the data distribution is given, and assuming that sampling longer sentences doesn't inherently change that data distribution, we can expect that as the number of samples grows, so does the density of the data, and therefore the size of the $\varepsilon$-neighborhoods all grows linearly with $b$.

Two small caveats to the above: 1. the $A_j$ aren't I.I.D along the sequence dimension - they're context embeddings which are by definition dependent. However, even if the LLM can exploit this to adversarially "mess up" PAMM's approximation, e.g., by placing many embeddings with virtually no neighbors, this doesn't happen in practice, as evident by our empirical success. 2. The claim above deals with $\mathbb{E}[N_\varepsilon(i)]$ and not $\mathbb{E}[\min_i |N_\varepsilon(i)|]$. However, the scaling of $\mathbb{E}[|N_\varepsilon(i)|]$ is the same for all $i$, even for the minimum so our claim holds for $n_{min}$ in particular.

# D    PRE-TRAINING

As mentioned in Section 4.2, we train LLaMA models (Touvron et al., 2023) with 60M-7B parameters on the C4 dataset (Raffel et al., 2023) (details for the 7B model are reported separately in Appendix E). We follow the training setup outlined in previous works (Zhao et al., 2024; Shamshoum et al., 2025). We train LLaMA-60M model for 10K steps, the LLaMA-350M for 60K steps and LLaMA-1B for 100K steps.

All runs use a maximum sequence length of 256 and a global batch size of 512. We adopt the scaling factor $\alpha = 0.25$ and tune the base learning rate $\eta \in \{1e-3, 3e-3, 5e-3, 8e-3, 1e-2\}$ separately for each model size. All weights are updated with $\eta$, except those trained with PAMM, which use the reduced rate $\tilde{\eta} = \alpha \cdot \eta$ for stability. Our learning rate scheduler employs a linear warm-up over the first 10% of the training steps, followed by a cosine decay that gradually reduces the learning rate to 10% of its maximum value over the remaining steps.

In Table 5, we can see the perplexity for different ranks with PAMM.

Table 5: **Pretraining perplexity and peak attention memory for different model sizes compared with the baseline.** Reported memory corresponds to the memory consumed by the activations of all the $Q, K, V$ projection layers. For PAMM measurements, this include the $\alpha$ and $f(\cdot)$. PAMM reduces the memory by more than 97% while gaining 0.56 perplexity for the LLaMA 1B model.

| Model size | 60M | | 350M | | 1B | |
|---|---|---|---|---|---|---|
| | **Perplexity** | **Memory** | **Perplexity** | **Memory** | **Perplexity** | **Memory** |
| Full Rank | **30.97** | 256 MB | 18.80 | 1.5 GB | 15.56 | 3 GB |
| PAMM $r = 1/128$ | $31.94 \pm 0.008$ | 8 MB | $\mathbf{18.40 \pm 0.04}$ | 42 MB | $\mathbf{15.01 \pm 0.04}$ | 78 MB |
| PAMM $r = 1/256$ | $32.18 \pm 0.03$ | 5 MB | $18.48 \pm 0.09$ | 24 MB | $15.06 \pm 0.02$ | 42 MB |
| PAMM $r = 1/512$ | $32.53 \pm 0.05$ | 3.5 MB | $18.49 \pm 0.03$ | 15 MB | $15.36 \pm 0.04$ | 24 MB |
| Training tokens | **1.1B** | | **6.4B** | | **13.1B** | |

## D.1    PAMM AND FLASHATTENTION

All experiments use FlashAttention-v2 (Dao, 2023), a faster and more memory-efficient implementation of the scaled–dot-product attention mechanism. FlashAttention avoids materialising the full attention-score matrix of size $L \times L$; instead, it stores only the operator's inputs and outputs together with a few small work buffers. In particular, the output of the scaled–dot-product operation is retained as part of FlashAttention's activation set. However, this tensor is also the input - and the activation tensor - of the linear output-projection layer. Namely, it is *shared* between FlashAttention and the output projection, being saved in memory only once.

Therefore, When considering PAMM compression for the output-projection layer, several options arise:

1) **Compress the shared activation itself.** While memory-efficient, altering this shared tensor perturbs the gradient flowing into FlashAttention, introducing error accumulation through layers and to the residual stream - an effect we do not study in this work.

2) **Compress a separate copy kept by the output-projection layer.** This *increases* memory usage, because the original (uncompressed) activation is still stored by FlashAttention.

3) **Remove FlashAttention completely so it does not retrain this tensor.** This *largely increases* memory usage, as the naive implementation of attention is far more memory intensive, even if we apply PAMM for the output projection layer.

Thus, to avoid memory increase or performance degradation, we leave the output-projection layer untouched and defer a joint treatment of FlashAttention and PAMM compression to future work.

# E    LLaMA-7B Results

We pretrained a LLaMA-7B model with and without PAMM, using $r = 1/256$ and $r = 1/512$, and report the models' perplexity in Table 6. The training setup follows previous works (Zhao et al., 2024). We train the model for 150K steps, with a global batch size of 512 and sequence length of 256, on 8 GPUs. A scaling factor $\alpha = 0.25$ was used with learning rate $\eta \in \{1e-3, 7e-4\}$.

|          | 40K   | 80K   | 120K  | 150K  |
|----------|-------|-------|-------|-------|
| PAMM-256 | 17.73 | 14.93 | 13.91 | 13.81 |
| PAMM-512 | 17.53 | 14.62 | 13.65 | 13.57 |
| Baseline | 18.09 | 15.47 | 14.83 | 14.61 |

Table 6: Pre-training LLaMA 7B on C4 dataset for 150K steps. We report perplexity with and without PAMM.

# F    LLaMA-1B Runtime Breakdown

We now report a breakdown of the runtime of the different operations in PAMM's forward and backward passes using a LLaMA-1B model, as well as the runtime of the same model without PAMM. We use the same batch size used in Section 4.4, and report time in ms, as well as the operation's relative contribution to the total iteration's runtime (forward + backward + optimizer step). We also report the relative contribution of the appropriate phase (forward or backward). Results are in Table 7 and Table 8.

| Operation          | Time (ms) | % of iteration | % of forward |
|--------------------|-----------|----------------|--------------|
| PAMM forward total | 101       | 6.7%           | 19.1%        |
| Forward pass matmul| 53        | 3.5%           | 10.0%        |
| Index selection    | 12        | 0.8%           | 2.3%         |
| Normalization      | 22        | 1.5%           | 4.2%         |
| Cosine matmul      | 8         | 0.5%           | 1.5%         |
| Max/assign         | 3         | 0.2%           | 0.6%         |

Table 7: PAMM forward pass breakdown for LLaMA-7B. Times are averaged over 1000 update steps. Percentages are reported relative to the full training iteration and to the total forward time.

| Operation           | Time (ms) | % of iteration | % of backward |
|---------------------|-----------|----------------|---------------|
| PAMM backward total | 149       | 9.9%           | 15.8%         |
| Input grad matmul   | 51        | 3.4%           | 5.4%          |
| Index gathering     | 21        | 1.4%           | 2.2%          |
| Alpha scaling       | 17        | 1.1%           | 1.8%          |
| Matmul              | 57        | 3.8%           | 6.0%          |

Table 8: PAMM backward pass breakdown for LLaMA-7B. Times are averaged over 1000 update steps. Percentages are reported relative to the full training iteration and to the total backward time.

We emphasize that the Argmax operation is implemented efficiently in parallel using tree reduction. The time of the random sampling (Index Selection) is negligible - less than 1% of the total forward time as shown in Table 7. We also experimented with reusing generators, but that produced no speed gain, and introduced small perplexity degradation. Therefore, we opted for the default choice of per-step sampling.

## G FINETUNING

**Evaluation** For the GLUE benchmark (Wang et al., 2019), the metrics reported are F1-score for MRPC, Matthew's correlation for CoLA, Pearson's correlation for STS-B, and accuracy for the rest of the tasks, along with the peak memory consumed by the activations of the three linear projections. Baseline scores are taken from CompAct (Shamshoum et al., 2025). For every task, we average results over three random seeds.

**Training Setup** We trained the various models for 30 epochs using batch size 16, except for CoLA where we used batch size 32. We perform a hyperparameter search once on RTE for the optimal PAMM scale factor $\alpha$ for two ranks, $r \in \{1/128, 1/256\}$. The sweep $\alpha \in \{1, 2, 4, 8\}$ yields $\alpha = 4$ for $r = 1/128$ and $\alpha = 2$ for $r = 1/256$; these settings are fixed for all other tasks. For the learning rate we consider $\{1e-5, 2e-5, 3e-5\}$ for every task, and consistently found that $lr = 1e-5$ worked best.

We note that for some tasks, such as CoLA it happens that $B \cdot L < 128$. Thus, following our method definition, such iterations used $k = \lceil r \cdot B \cdot L \rceil$. These provide empirical evidence for the effectiveness of PAMM even with an extreme $k = 1$.

## H PAMM COMPRESSION EDA

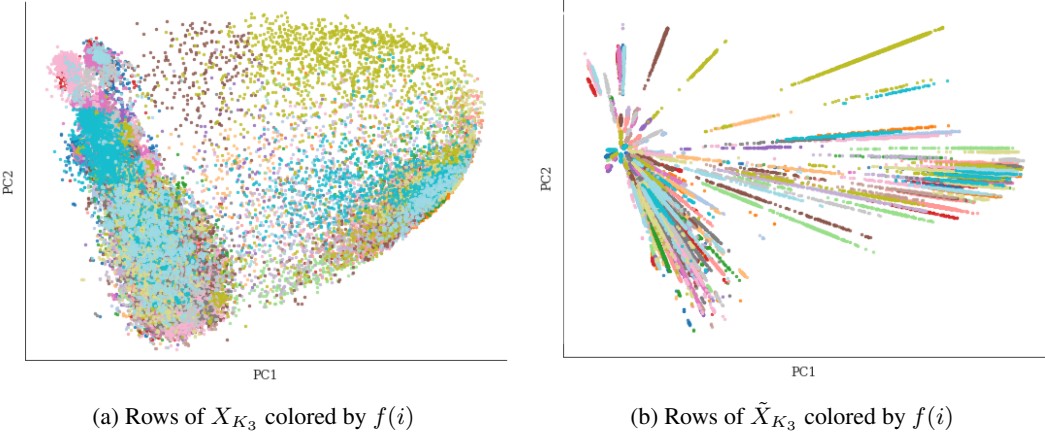

(a) Rows of $X_{K_3}$ colored by $f(i)$      (b) Rows of $\tilde{X}_{K_3}$ colored by $f(i)$

Figure 5: **Two-dimensional PCA visualization of PAMM approximation.** **(a)** rows of the input tensor of the $K$ projection of layer 3 in LLaMA-60M, $X_{K_3}$. The rows are projected using PCA and colored by their assigned generating point, i.e. by $f(i)$, using PAMM with $\varepsilon = \infty$. **(b)** The representative rows $\tilde{X}_{K_3}$ shown in the same PCA space, colored by $f(i)$. We can see that our method approximately clusters the input data, and transforms clusters into lines.

In this appendix we examine the behaviour of PAMM on real data taken from one of our LLaMA-60M pretraining experiments, with $\varepsilon = \infty$. We chose to use the input activation to the projection $K$ at layer 3, at training step 3000. We write $X_{K_3} \in \mathbb{R}^{b \times n}$ in this section to denote this data tensor. For LLaMa-60M, $b = 65536, n = 512$.

In Figure 5 we plot the first 2 principle components of the PCA decomposition of $K_3$. We run the first stage of PAMM, selecting $k$ generating points $\{C_j\}_{j=1}^k$, and color each point in $X_{K_3}$ according to its chosen generator, i.e. according to $f(i)$. Next, we plot the PCA of the representative points $\tilde{K}_3$, using the same PCA projection from the first step. We use the same coloring scheme. As can be seen, PAMM's compression approximately clusters the input data; close points share the same generator. We also see that the representatives of each cluster lie on the same line, which is expected from our construction of $\tilde{A}_i = \alpha_i C_{f(i)}$. Finally, notice that the variation in the first two principle components is mostly preserved for the whole set of points, while each cluster's variance is greatly reduced. Since our perplexity results were very good, this indicates that indeed most of the lost variance was redundant.

Next, we measure the estimation error of PAMM for the same layer. In addition to the input tensor $X_{K_3}$, we also look at $\nabla K_3$, the gradient of the projection $K$ recorded during the backward step of the same training iteration. We compute the exact weight gradient $\nabla W_{K_3} = X_{K_3}^\top \cdot \nabla K_3$, and also compute the PAMM approximation $\widetilde{\nabla W}_{K_3} = \tilde{X}_{K_3}^\top \cdot \nabla K_3$. To assess how well PAMM approximates matrix multiplication, we measure the relative $L_2$ error:

$$E(r, \varepsilon) = \frac{\|\nabla W_{K_3} - \widetilde{\nabla W}_{K_3}\|_F}{\|\nabla W_{K_3}\|_F}$$

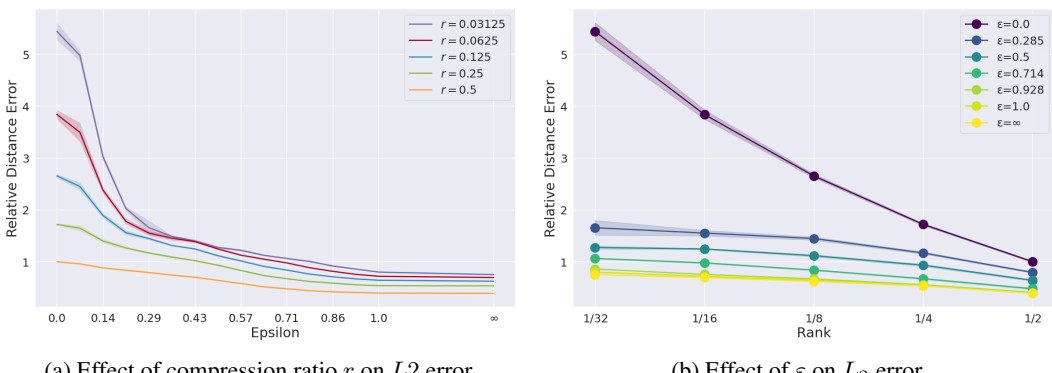

(a) Effect of compression ratio $r$ on $L2$ error.    (b) Effect of $\varepsilon$ on $L_2$ error.

Figure 6: **Relative $L_2$ error analysis.** We compare the relative $L_2$ error $E(r, \varepsilon)$ for multiple compression ratios and with different epsilon values, including $\varepsilon = \infty$ which was used in our main experiments. The relative $L_2$ errors are calculated on the parameter gradients $\nabla W_{K_3}$ of LLaMa-60M model after 3K training steps. We can clearly see that the error decreases as $\varepsilon$ increases, indicating that $\varepsilon = \infty$ is the best choice of this parameter. We also observe that the errors scale only logarithmically with $r$, which explains why we could reduce $r$ so drastically in our experiments.

Notice that $E$ depends on our selection of compression ratio $r$ and neighborhood parameter $\varepsilon$. We repeat this measurement for various values of $r$ and $\varepsilon$ and present the results in Figure 6. Additionally, we also measure the coverage of PAMM for each $\varepsilon, r$, and present the results in Figure 7

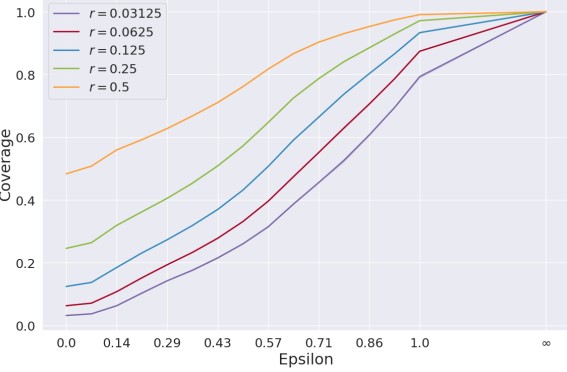

Figure 7: **PAMM Coverage**: Coverage is the ratio between the number of points that have a representative, i.e. their $\alpha_i \neq 0$, to the total number of points in the tensor. Coverage is reported for several compression ratios $r$ and epsilon values $\varepsilon$. As expected, coverage rises with larger $\varepsilon$ values, as more points satisfy the neighborhood criterion (Equation 2). Likewise, higher $r$ values increase coverage by introducing more generator points, increasing the number of rows for which there is a generator satisfying the neighborhood condition, and representing a greater portion of the data.

We have several key insights from the results. First, as $\varepsilon$ increases, and more points are covered by the approximation (Figure 7), the relative error drops (Figure 6a). This further supports our results

from 4.6 which indicate that choosing $\varepsilon = \infty$ is the best option. Second, we see that as the relative error scales only logarithmically with $r$ (Figure 6b). This shows that $r$ can indeed become very very small without significantly reducing performance, as the error scales up relatively slowly. Third, for $\varepsilon = 0$, i.e. for Uniform-CRS, the error is much higher compared to $\varepsilon = 0.2$. This further strengthens our results from Section 4.6 where we show PAMM outperforms Uniform-CRS.

Lastly, we note that the relative errors we measured, even for $\varepsilon = \infty$, are on the order of $1/2$ to $1$. These are large relative error values; this means that for the data distribution present in real transformers, PAMM isn't a very good approximation in terms of relative norm error. However, the perplexity we achieved in practice was very close to the baseline, indicating again that the training process of LLMs is highly redundant. Even though PAMM doesn't approximate the full gradient very well, the model is still perfectly able to learn as well as without any compression.

## I  LOSS CURVES

Figure 8 compares the pretraining loss curves of LLaMa-1B with and without PAMM to verify the stability of our method and its effect on training dynamics. We can see the resulting curves exhibit nearly identical behavior, and that PAMM maintains smooth and stable training, asserting that PAMM does not negatively affect optimization. This stability was consistent across all three random seeds.

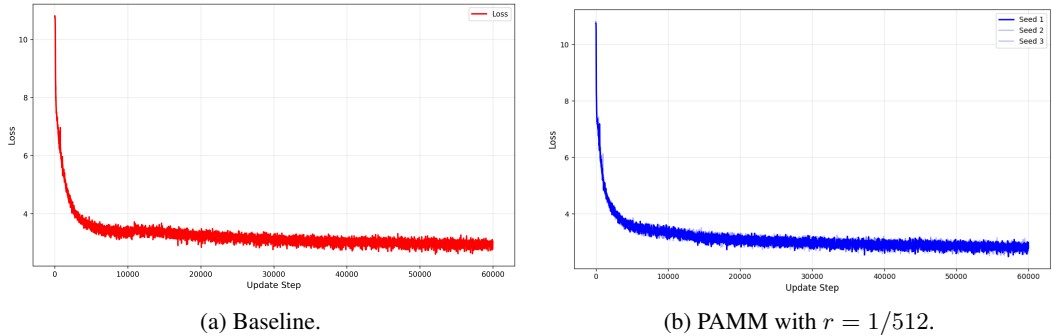

(a) Baseline.                                    (b) PAMM with $r = 1/512$.

Figure 8: **Pretraining Loss Curve Comparison.** We validate PAMM's stability by evaluating it on multiple seeds and comparing it to the baseline LLaMa-1B pretraining in the same setting. The loss is equivalent to model perplexity on the C4 dataset.

## J  COMPLEXITY ANALYSIS

We now compare the memory and time complexity of PAMM with the exact matmul operation.

**Memory Complexity**  The original matrix $A$ requires storing $bn$ scalars. PAMM only uses $kn + 2b = O(kn)$ scalars to represent $\tilde{A}$, made up of the matrix $C \in \mathbb{R}^{k \times n}$ and the two vectors $f \in [k]^b$ and $\alpha \in \mathbb{R}^b$. We show that in practice $k \ll b$, reducing the total memory required to represent $\tilde{A}$ to a very small fraction of the original memory.

**Time Complexity**  The number of scalar multiplications in the full matrix multiplication $O = A^\top B$ is $bnm$. During compression, PAMM computes the scalar product of every $A_i$ and $C_j$, i.e. we compute $AC^\top$ which has $bkn$ multiplications. We also compute the squared norms of $A_i$ which require an additional $bn$ multiplications. Finally we perform $b$ $\arg\max$ operations which can be efficiently computed using parallel reduction algorithms in $b \cdot log(k)$ time. For decompression, in order to compute $\tilde{B}$ we have $bm$ multiplications, and to compute $\tilde{O} = C^\top \tilde{B}$ we have $knm$ multiplications. In total, the number of multiplications used in PAMM is $O(bkn + knm + bn + bm + b \cdot log(k)) = O(kn(b+m))$ Comparing this with the original $bnm$ we see that the speedup ratio $\gamma = \frac{bm}{k(b+m)}$ determines if PAMM will be faster or slower than exact multiplication: if $\gamma > 1$, PAMM will be faster. This speedup is highly dependent on model size and parameter choice of $k$, and in our

experiments, we had $\gamma$ values as high as $\approx 28$ for LLaMa-1B during pretraining, using $k = b/256$. However, the matmul operation PAMM is used to approximate in our work is only a small part of the entire runtime of the model, so the total training runtime is negligible for large models. Also, the theoretical speedup doesn't account for implementation details, including memory movement in the GPU between the GPU's RAM and the SMs doing the actual computation, as well as adding several CUDA kernel launches. In practice, as model size increases, the total overhead incurred by PAMM becomes negligible. We give a detailed analysis of training throughput in Section 4.4.

