# OpenReview forum: "QKV Projections Require a Fraction of Their Memory"
_ICLR.cc/2026/Conference — ICLR 2026 Poster_

### Official Review · Reviewer_16on · 2025-10-30

**Soundness:** 3
**Presentation:** 3
**Contribution:** 2
**Rating:** 4
**Confidence:** 5

**Summary:**

This paper proposes Point-Approximate Matrix Multiplication (PAMM) to reduce memory consumption of the QKV projections, by a factor of up to 512x, thereby improving memory footprint while achieving similar PPL. PAMM can be composable with other efficient attention techniques such as FlashAttention and LoRA, making it practical and complementary for memory-efficient LLM training.

**Strengths:**

1. This paper shows that QKV projections are highly redundant along the sequence dimension and thus can be compressed by a factor of >100x for efficient training with almost no accuracy loss, marking a step toward large-size model training on limited hardware devices.
2. The throughput degradation decreases when the model size gets larger, making PAMM more practical for large-size model training on limited hardware devices -- highly reduced memory consumption (hard constraint, must be satisfied) and low throughput degradation (soft constraint, the lower the better).

**Weaknesses:**

1. The memory consumption of QKV projections is significantly reduced, but that is just the memory consumption of QKV projections and accounts for 20% of peak usage -- there are many other tensors/activations consuming non-negligible memory. Therefore, the overall impact of this work may be limited (by 20%).
2. The throughput degradation is proportional to batch_size * seq_len -- but only the effect of model size on the throughput degradation is discussed in the paper. The proposed method may be (much) less efficient with large batch size or long sequence length -- this should also be addressed.
3. The largest model used in the experiments is LLaMA-1B -- larger model(s) should be considered.

**Questions:**

My questions and suggestions are basically from "Weaknesses" as aforementioned.
1. From Weakness 1: If my opinion is correct, PAMM can save up to (and at most) 20% of peak memory usage.
2. From Weakness 2: Please discuss the effects of batch size and sequence length on the throughput degradation.
3. From Weakness 3: Please use larger model(s) to demonstrate the consistency of PAMM's efficacy.

---

> ### Author Response · Authors · 2025-11-25
> **Response to Reviewer 16on**
>
> We thank the reviewer for their careful reading and for highlighting both PAMM’s strong memory-performance tradeoff and its practicality for large-model training on limited hardware.
>
> Responding to your questions:
> 1. **QKV Memory Consumption**
> The total memory of training an LLM is dominated by activation tensors saved for backwards, but the attention activation tensors, which PAMM targets, take a relatively small part of this total. The main insight of our work is that although end-to-end memory includes many components, the attention activation memory can be almost fully eliminated using PAMM without any loss. This is a surprising result with implications for our current understanding of the attention mechanism.
> We will add a detailed breakdown of the total memory consumption including other layers as an appendix.
> We also plan on exploring ways to apply PAMM to other layers of the LLM in future work, which would be complementary to our current results.
>
> 2. **PAMM’s Throughput**
> The algebra of PAMM flattens the batch and sequence dimensions together. So in principal as long as the multiplication $b=BL$ is constant, we expect the same throughput and computational costs for PAMM. As for how this throughput scales for longer sequences or larger batch size, we ran an ablation on PAMM computational cost for LLaMA-1B with different batch sizes and sequence lengths, and display the result in the table below. The results show a similar computational cost ratio compared to the baseline across the different batch size and sequence length values.
> We will add this ablation, as well as a discussion of the results to the revised paper.
>
>      | Batch Size (per GPU) | Sequence Length | Method   | Total Iter (ms / 1k tokens) | Forward (ms / 1k tokens) | Backward (ms / 1k tokens) | Δ total vs. Baseline (%) |
>      |------------|------|----------|-------|----------|-----------|-----------|
>      | 64     | 256   | Baseline | 88.58     | 31.53    | 57.05     | -      |
>      | 64     | 256   | PAMM     | 92.21     | 33.15    | 59.06     | +4.09%       |
>      | 32     | 512   | Baseline | 88.87      | 31.89    | 54.78     | -      |
>      | 32     | 512   | PAMM     | 93.10      | 34.04    | 56.92     | +4.75       |
>      | 16     | 1024  | Baseline | 89.91      | 32.21    | 55.50     | -      |
>      | 16     | 1024  | PAMM     | 94.51      | 34.26    | 58.10     | +5.11       |
>
> 3. **Scaling PAMM to larger models**
> To address the reviewer’s concerns regarding scaling PAMM to larger models and longer sequence lengths, we ran several experiments added below. The results all indicate that PAMM scales well to larger models, different sequence lengths and different batch sizes.
>
>     * **Pretraining LLaMA-7B**
>     During the rebuttal period, we pretrained **LLaMA-7B** on C4 with PAMM. Results match our paper’s trend: **slightly improved perplexity (5.47%)** together with the **aggressive x256 QKV activation compression**, confirming PAMM scales well to a larger scale.
>     In detail, PAMM achieves a perplexity of 13.91 compared to 14.83 perplexity of the baseline after 120K iterations of training. We report here the perplexity of the model vs. the baseline’s perplexity (full rank training), against sample iteration counts. The table will also be added to the paper body.
>     We are also running a similar model with x512 compression which is not yet finished.
>
>          |   | 40K  | 80K  | 120K | 150K |
>          |-|-|-|-|-|
>          | PAMM    | 17.73| 14.93| 13.91| 13.81|
>          | Baseline| 18.09| 15.47| 14.83| 14.61|
>
>     * **Finetuning Multi-modal Pixtral-12B Vision Language Model (VLM)**
>     Additionally, we finetuned **Pixtral-12B VLM** with LoRA+PAMM on the AID satellite image classification benchmark [1,2].
>     Specifically, pretrained Pixtral-12B achieves an F1 score of 0.56 (Averaged over 30 classes), while LoRA finetuning achieves 0.97. LoRA+PAMM finetuning achieves the same 0.97 F1 score, while also saving 453MB of GPU memory.
>     This result shows that:
>         * PAMM shows consistent results for even **larger models - 12B parameters**.
>         * PAMM is applicable to multi-modal **VLMs**.
>         * PAMM is compatible with common **PEFT** methods such as LoRA.
>
>     We will add these results to the paper with full explanations of the training setup.
>
>
> We hope these clarifications have answered your questions, and we’re happy to answer more if needed. If you’re satisfied with the above answers please consider improving our scores.

---

### Official Review · Reviewer_i9xN · 2025-10-31

**Soundness:** 2
**Presentation:** 2
**Contribution:** 2
**Rating:** 6
**Confidence:** 3

**Summary:**

PAMM proposes to approximate GEMMs using a sequence of point-wise multiplications. These are obtained by projecting one of the GEMM operands into lines spanned by a small catalog of points randomly sampled from the rows of said operand.  Each point in the activation tensor is represented by its closet point from a generator, under a proposed "neighborhood condition".

**Strengths:**

- The proposed method identifies the redundancy in the sequence dimension, which is typically enormous in modern LLM training.
- Outperforms other methods like CompAct and Uniform-CRS in memory-performance tradeoffs.

**Weaknesses:**

- the experiments are limited to small to medium size LLMs. Scaling up models can introduce performance degradation of the approximation method.
- Activations are dynamic during training. Random sampling is slow, particularly on GPUs. Note that random sampling requires a pseudo-random number generator, such as an LFSR or others. Most algorithms in this family are sequential and model generation from an irreducible in a Galois field. The authors should check the intrinsic details of cudaranddx. Can we just sample once and stick with the choice of catalog points Cjs, or perhaps periodically refresh?
- The algorithm requires a matmul followed by an argmin on one of the dimensions. This means a reduction over one of the reduction, this is slow on a GPU.
- The breakage of computation into a sequence of small GEMMs can harm tensor core utilization.

**Questions:**

No questions, but please address the above.

---

> ### Comment · Reviewer_i9xN · 2025-11-25
> **No rebuttal**
>
> The authors have not provided a rebuttal. Granted, I had given a score of 6; but since I received no response, I will not champion this paper.

---

> > ### Author Response · Authors · 2025-11-25
> >
> > We ran a lot of experiments to address your concerns, and prepared detailed responses. We will upload the response in the next few hours.

---

> ### Author Response · Authors · 2025-11-25
> **Response to Reviewer i9xN (Part 1/2)**
>
> We thank the reviewer for the constructive comments and for highlighting PAMM’s ability to exploit sequence-axis redundancy and outperform existing baselines.
>
> 1. **Scaling PAMM to larger models**
> To address the reviewer’s concerns regarding scaling PAMM to larger models, we ran several experiments added below. The results indicate that PAMM scales well to larger models.
>
>     * **Pretraining LLaMA-7B**
>     During the rebuttal period, we pretrained **LLaMA-7B** on C4 with PAMM. Results match our paper’s trend: **slightly improved perplexity (5.47%)** together with the **aggressive x256 QKV activation compression**, confirming stable behavior at larger scale. Specifically, PAMM achieves a perplexity of 13.91 compared to 14.83 perplexity of the baseline after 120K iterations of training. We report here the perplexity of the model vs. the baseline’s perplexity (full rank training), against sample iteration counts. The table will also be added to the paper body.
>     We are also running a similar model with x512 compression which is not yet finished.
>
>         |    | 40K  | 80K  | 120K | 150K |
>         |-----|---|---|---|---|
>         | PAMM    | 17.73| 14.93| 13.91| 13.81|
>         | Baseline| 18.09| 15.47| 14.83| 14.61|
>
>
>     * **Finetuning Pixtral-12B**
>      Additionally, we finetuned **Pixtral-12B (VLM)** with LoRA+PAMM on the AID satellite image classification benchmark [1,2].
>      Specifically, pretrained Pixtral-12B achieves an F1 score of 0.56 (Averaged over 30 classes), while LoRA finetuning achieves 0.97. LoRA+PAMM finetuning achieves the same 0.97 F1 score, while also saving 453MB of GPU memory.
>     This result shows that:
>         * PAMM shows consistent results for even **larger models** - 12B parameters.
>         * PAMM is applicable to multi-modal **VLMs**.
>         * PAMM is compatible with common **PEFT** methods such as LoRA.
>
>      We will add these results to the paper with full explanations of the training setup.
>
>     [1] Agrawal, Pravesh, et al. “**Pixtral 12B**”. 2024. arXiv:2410.07073.
>     [2] Xia, Gui-Song, et al. “**AID**: A Benchmark Data Set for Performance Evaluation of Aerial Scene Classification.” IEEE Transactions on Geoscience and Remote Sensing, vol. 55, no. 7, 2017, pp. 3965–3981, doi:10.1109/TGRS.2017.2685945.
>
> 2. **Cost of random sampling on GPUs:**
> The computational cost of PAMM’s random sampling step is negligible relative to the rest of the linear layer. Random operations on GPUs are standard and heavily used in practice (e.g., for dropout, diffusion models), so this component does not introduce any unusual overhead. We also experimented with reusing the same generators across multiple steps (and refreshing them periodically). This change produced no measurable throughput improvement (the generator sampling itself is already very cheap), but it did cause a small degradation in final perplexity for small models compared to sampling new indices at every step. We therefore adopt per-step sampling in all our experiments.
>      _
>     To emphasize how small this effect is in practice, we give below a **breakdown** of the training iteration of PAMM’s major operations, and compare them with the total iteration time which is the (forward+backward+optimizer step) time. Measurements were done with LLaMA-1B in with batch size 64 per GPU and sequence length 256, and averaged over 1000 iterations. The results show that sampling the generators (Index Selection) takes only 0.8% of the full iteration time, so replacing it with something more efficient is simply unnecessary. The full PAMM algorithm adds 4% to the runtime in this configuration.
>
> ```text
> =======================================================
> PAMM LLaMA-1B Full Iteration Breakdown
> =======================================================
> Total iteration time:    1509 ms (100.0%)
>   ├─ Model forward:  530 ms ( 35.1% of total)
>   │   └─ PAMM (96 layers):  101 ms ( 6.6% of total)
>   ├─ Model backward: 944 ms ( 62.5% of total)
>   │   └─ PAMM (96 layers):  149 ms ( 9.8% of total)
>   └─ Optimizer step:  35 ms (  2.3% of total)
> =======================================================
>
> =======================================================
> Baseline LLaMA-1B Full Iteration Breakdown
> =======================================================
> Total iteration time:    1448 ms (100.0%)
>   ├─ Model forward:  518 ms ( 35.8% of total)
>   ├─ Model backward: 894 ms ( 61.8% of total)
>   └─ Optimizer step:  36 ms ( 2.5% of total)
> =======================================================
>  ```
> Please see the rest of our response below.

---

> ### Author Response · Authors · 2025-11-25
> **Response to Reviewer i9xN (Part 2/2)**
>
> ```text
> =======================================================
> PAMM LLaMA-1B Forward Pass Timing
> =======================================================
> Operation breakdown:
>   ├─ Forward pass matmul:    53 ms ( 3.5% of total)
>   └─ PAMM compression:  51 ms ( 3.4% of total)
>     ├─ Index selection:      12 ms ( 0.8% of total)
>     ├─ Normalization:   22 ms ( 1.5% of total)
>     ├─ Cosine matmul:      8 ms ( 0.5% of total)
>     └─ Max/assign:       3 ms (  0.2% of total)
> =======================================================
>
> =======================================================
> PAMM LLaMA-1B Backward Pass Timing
> =======================================================
> Operation breakdown:
>   ├─ Input grad matmul:     51.07ms (3.4% of total)
>   └─ PAMM decompression (param grad):    94.54 ms (6.3% of total)
>     ├─ Index gathering:    21.40ms (1.4% of total)
>     ├─ Alpha scaling:  16.60ms (1.1%  of total)
>     └─ Matmul:        56.54ms (3.7% of total)
> =======================================================
> ```
> 3. **Slow operations on GPU:**
> The argmax operation can be efficiently implemented in parallel in $O(log(n))$ steps using tree reduction techniques. On the GPU these are typically implemented by having each thread warp in the GPU compute a warp-level argmax using warp-level operations such as shift, and then the global max is computed over shared memory or on-device cache.
> This sort of implementation is standard practice and is the default in pytorch’s wrapped libraries. Additionally, the final matrix multiplication is smaller than the original MM as the dimension is significantly reduced.
>
>
> 4. **PAMM is not a sequence of small GEMMs**
> PAMM doesn’t break the computation down into small matmuls.
> As shown in Section 3 as well as in Appendix A, we sample the generators in parallel; we compute the projection coefficients using one large matmul ($C \cdot A$); we compute the argmax using a fast tree-reduction algorithm (wrapped in pytorch’s linked libraries); in the backward pass we sum rows of B after multiplication with the appropriate projection coefficient using an index_add operation; and finally we compute ($C \cdot B$) which is one GEMM.
> In total, one very large GEMM was replaced with two smaller GEMMs.
> Also, even if this utilizes the GPU less optimally, PAMM is fast enough to have negligible effect on the total iteration runtime.
>
> We hope these clarifications have answered your concerns, and we’re happy to answer more if needed. If you’re satisfied with the above answers please consider improving our scores.

---

> > ### Comment · Reviewer_i9xN · 2025-11-26
> > **Thanks**
> >
> > Thanks for the responses. Is the profiling done using nsys?

---

> > > ### Author Response · Authors · 2025-11-27
> > > **Profiling**
> > >
> > > The profiling was done using cuda events and syncs, which is sufficient for timing the breakdown we attached above. See precise pseudocode below for total forward time as an example:
> > >
> > > ```text
> > > start = torch.cuda.Event(enable_timing=True)
> > > end = torch.cuda.Event(enable_timing=True)
> > > for _ in n_iters:
> > >     synchronize()
> > >     start.record()
> > >     out = model()
> > >     end.record()
> > >     synchronize()
> > >     average_forward += start.elapsed_time(end)/n_iters
> > > ```

---

> > > > ### Comment · Reviewer_i9xN · 2025-11-27
> > > > **Bug**
> > > >
> > > > end.record() should be after the second synchronize() otherwise you measure the elapsed time on the host only.
> > > >
> > > > I suggest using nsys to get a detailed breakdown of the elapsed times of each cuda kernel on the device.

---

> > > > > ### Author Response · Authors · 2025-11-27
> > > > > **Clarification on Profiling Methodology**
> > > > >
> > > > > Our implementation follows the standard PyTorch and CUDA programming model for asynchronous timing.
> > > > >
> > > > > Unlike CPU timers (e.g., time.time()), torch.cuda.Event.record() does not record the host timestamp immediately. Instead, it places a marker in the GPU command stream. The PyTorch documentation explicitly states that the event is recorded 'after all preceding operations in the stream have been completed'.
> > > > >
> > > > > Therefore, in our code:
> > > > > ```
> > > > > model()          # 1. Enqueues the forward pass
> > > > > end.record()     # 2. Enqueues the timestamp marker immediately after
> > > > > synchronize()    # 3. Blocks Host until #1 and #2 are done
> > > > > ```
> > > > >
> > > > > This ensures end.record() captures the exact moment the GPU finishes the forward pass.
> > > > > If we were to move end.record() to after the synchronize() call (as suggested), we would be forcing the CPU to wake up, process the line, and submit a new command. This would introduce CPU wake-up latency and kernel launch overhead in the measurement, rather than measuring pure GPU execution time.
> > > > >
> > > > > Since our goal was to measure the GPU execution time of the method(forward in our example), the current placement provides the most accurate result without the need for full nsys profiling for this specific metric.

---

> > > > > > ### Comment · Reviewer_i9xN · 2025-11-27
> > > > > > **This assumes that only one stream is used on the GPU**
> > > > > >
> > > > > > There is no guarantee that pytorch does not use multiple streams. See here:
> > > > > > https://developer.download.nvidia.com/CUDA/training/StreamsAndConcurrencyWebinar.pdf
> > > > > >
> > > > > > Generally, it's fine to include the tiny amount of time the synchronize() function takes in order to ensure concurrency with the timing. You would think of it as a tight upper bound on what you are trying to measure.
> > > > > >
> > > > > > Alternatively, you can use a profiler such as nsys to get a breakdown of all streams and events.

---

> > > > > > > ### Author Response · Authors · 2025-12-01
> > > > > > > **Throughput Measurement is Correct**
> > > > > > >
> > > > > > > We’ve clarified that our implementation relies on the standard PyTorch CUDA execution model. We do not explicitly create side streams for the measured compute regions. According to PyTorch documentation, all operations on a device run on the "default stream" unless specified otherwise, and the backward pass executes on the same stream to ensure consistency [1].
> > > > > > >
> > > > > > > To fully address the concern, we also repeated all profiling using the reviewer’s suggested sync-before-record approach. Both methods produced almost identical results (<0.2% difference), so the reported performance and **our conclusions remain unchanged.**
> > > > > > >
> > > > > > > References: [1] PyTorch Documentation, "CUDA Semantics - Default Stream": https://pytorch.org/docs/stable/notes/cuda.html

---

### Official Review · Reviewer_qFLe · 2025-10-31

**Soundness:** 2
**Presentation:** 4
**Contribution:** 2
**Rating:** 4
**Confidence:** 5

**Summary:**

This paper tackles the less explored problem of reducing activation memory during LLM's training. The authors propose Point-Approximate Matrix Multiplication (PAMM), a simple technique that compresses the input activations of the Q, K, and V projections in attention layers using a small set of representative rows and scaling factors. This allows approximate gradient computation without storing full activations. Experiments on LLaMA models (60M–1B) and RoBERTa-base show that PAMM reduces activation memory by $512 \times$ with little to no loss—and sometimes improvement—in perplexity and task accuracy. The method introduces only a small training throughput penalty and is compatible with existing efficiency techniques.

**Strengths:**

* The paper is well written, clearly structured, and easy to follow.
* The proposed method is supported by solid theoretical analysis.
* The approach effectively reduces training-time memory consumption while maintaining, or even slightly improving, model accuracy with minimal degradation.

**Weaknesses:**

* **Limited profiling on sequence redundancy:** The paper offers initial empirical evidence of sequence‑axis redundancy (Appendix F: PCA clustering; relative error and coverage), but the analysis is confined to a narrow slice (one layer/model/step). A broader study would better ground the motivation.

* **Missing complexity/scaling analysis:** While the runtime breakdown is helpful (Table 2), the paper lacks an explicit complexity analysis of the compression and approximate multiply and a scaling study with respect to batch size, sequence length, and model dimension.

* **Narrow evaluation scope:** Pretraining results are limited to a specific setup, with sequence length 256 and batch size 512. It is unclear how PAMM performs under different training regimes (e.g., smaller batch but longer sequences).

* **End‑to‑end memory not reported:** Memory savings are reported for QKV activations only. End‑to‑end peak memory (including other activations, parameters, and optimizer states) and a memory timeline would better quantify the overall training benefit.

* **Throughput comparability:** Throughput is reported for single‑GPU batches of 16K tokens (Section 4.4), whereas pretraining uses a global batch of 128K tokens (Appendix D). Aligning these settings or clarifying per‑GPU vs. global configurations would strengthen the runtime claims.

**Questions:**

* Do similar sequence-level redundancies exist in pretrained LLMs during inference. Does the degree of redundancy change over the course of training?

* Could the authors report the total training-time memory consumption (including all activations, parameters, and optimizer states) for both the baseline and the PAMM-compressed models?

* The current experiments use relatively short sequences (256 tokens) and moderate batch sizes. How does PAMM’s computational cost scale as sequence length or batch size increases?

* How sensitive is PAMM to the shape of the training batch—for example, using fewer long sequences versus more short ones with the same total number of tokens? Does this affect model accuracy or compression effectiveness?

* Can PAMM be applied to other activation tensors beyond the QKV projections, such as feed-forward or output-projection layers? How much redundancy do those tensors exhibit in comparison?

---

> ### Author Response · Authors · 2025-11-25
> **Response to Reviewer qFLe (Part 1/2)**
>
> We thank the reviewer for the thoughtful and detailed feedback, as well as for recognizing the clarity of our presentation, the practical relevance of the problem, and the strength of both our empirical and theoretical analysis. Below we address each of the raised concerns and questions.
>
>
> Responding to your questions:
> 1. **Redundancy during inference:**
> PAMM can indeed be applied during inference as well, and we’re planning to work on a follow-up paper which will explore this idea.
>     **Change of redundancy over training:**
> The goal of our paper is to exploit redundancy constructively. Our implementation uses a fixed compression ratio, which already indicates substantial redundancy in QKV activations, at least as large as the maximal compression rate we evaluated (x512!). Directly quantifying this redundancy is challenging: PAMM introduces non-negligible reconstruction error, yet training remains stable, so approximation error alone is not a reliable proxy for redundancy. Simply exploring dynamic schedules (e.g., gradually increasing compression over training), would require an impractically large sweep of experiments. A separate research work is required to directly answer this question.
>
>     Our results therefore provide strong empirical evidence of high redundancy, and a full characterization of its dynamics is an important direction for future work.
>
> 2. **End-to-end memory consumption:**
> We will add a report of the total training-time memory consumption as an appendix.
> The total memory of training an LLM is dominated by activation tensors saved for backwards, but the attention activation tensors, which PAMM targets, take a relatively small part of this total. The main insight of our work is that although end-to-end memory includes many components, the attention activation memory can be almost fully eliminated using PAMM without any loss. This is a surprising result with implications for our current understanding of the attention mechanism.
>
> 3. **Effect of sequence length or batch size on computational cost:**
> The algebra of PAMM flattens the batch and sequence dimensions together. So in principal as long as the multiplication $b=BL$ is constant, we expect the same throughput and computational costs for PAMM. We ran an ablation of PAMM throughput for LLaMA-1B with different batch sizes and sequence lengths, and report the result in the table below. These results show a similar computational cost ratio compared to the baseline across the different batch size and sequence length values.
> We will add this ablation, as well as a discussion of the results, to the paper.
>
>       | Batch Size (per GPU) | Sequence Length | Method   | Total Iter (ms / 1k tokens) | Forward (ms / 1k tokens) | Backward (ms / 1k tokens) | Δ total vs. Baseline (%) |
>       |-|-|-|-|-|-|-|
>       | 64| 256| Baseline| 88.58| 31.53| 57.05| -|
>       | 64| 256| PAMM| 92.21| 33.15| 59.06| +4.09%|
>       | 32| 512| Baseline | 88.87| 31.89| 54.78| -|
>       | 32| 512| PAMM| 93.10| 34.04| 56.92| +4.75|
>       | 16| 1024| Baseline | 89.91| 32.21| 55.50| -|
>       | 16| 1024| PAMM| 94.51| 34.26| 58.10| +5.11|
>
>   4. **Effect of Training shape on PAMM compression:**
> Using PAMM, we pretrained LLaMA-60M with different sequence lengths and batch sizes. The results below show that PAMM achieves comparable perplexity to the baseline model, indicating PAMM works well regardless of the choice of sequence length, batch size, or batch token count.
>
>       * **Impact of Batch Size on Training Performance**
>       | **Batch Size**| **Sequence Length**| **Baseline Perplexity ↓** | **PAMM Perplexity ↓** |
>       |-|-|-|-|
>       | 128| 256| 42.63| 43.01|
>       | 256| 256| 37.61| 37.25|
>       | 512| 256| 30.97| 32.46|
>
>       * **Impact of Sequence Length on Training Performance**
>       | **Batch Size** | **Sequence Length** | **Baseline Perplexity ↓** | **PAMM Perplexity ↓** |
>       |-|-|-|-|
>       | 512 | 128 | 37.38| 36.43|
>       | 512 | 256 | 30.97| 32.46|
>       | 512 | 512 | 29.16| 30.30|
>
>       * **Training Performance for constant token count in the batch (128K tokens)**
>       | **Batch Size** | **Sequence Length** | **Baseline Perplexity ↓** | **PAMM Perplexity ↓** |
>       |-|-|-|-|
>       | 256| 512| 33.32| 33.73|
>       | 128| 1024| 37.47| 37.03|
>
> 5. **Applying PAMM to different activations:**
> PAMM cannot be applied as is to activations of other layers, including MLP layers. Initial experiments show that significant modifications are required. Adapting to other layers is a natural next step and part of our planned follow-up work.
>
> Please see the rest of our response bellow.

---

> ### Author Response · Authors · 2025-11-25
> **Response to Reviewer qFLe (Part 2/2)**
>
> Responding to other points raised in the weaknesses section:
> 1. **Profiling Sequence Redundancy** - See above our response to your first question.
> 2. **Complexity/Scaling analysis**
> See above our response with throughput analysis for different sequence lengths and batch sizes.
> We also derived the asymptotic complexity of PAMM but omitted it from the paper because PAMM’s practical runtime is dominated by GPU memory access patterns rather than asymptotic FLOPs. We’ll add the analysis back to the paper, and also add it to the bottom of this comment for the reviewer’s convenience.
> The result is that PAMM has time complexity $O(rbnm+rnb^2)$ compared with the full matmul’s $O(bnm)$, with $r=1/256$ or $r=1/512$.
> 3. **Evaluation Scope** - See the ablation above.
> 4. **End-to-End Memory** - See answer to the corresponding question.
> 5. **Throughput Comparability** -
> All throughput measurements were reported with the exact same per-GPU settings, which means they are all comparable with each other.
> In all experiments, we used a global batch size of 512 and sequence length of 256. For LLaMA-1B model, training was distributed over 8 GPUs, so each GPU processed a local batch of 64. Since throughput was measured on LLaMA-1B, the reported numbers correspond to 16K tokens per GPU (64 x 256 tokens), with all GPUs running in parallel.
> We will add this explanation to the paper for better clarity.
>
> **PAMM complexity analysis**
> In terms of memory complexity, the original matrix $A$ requires storing $bn$ scalars. PAMM only uses $kn+2b=O(kn)$ scalars to represent $\tilde{A}$, made up of the matrix $C\in\mathbb{R}^{k \times n}$ and the two vectors $f\in[k]^b$ and $\alpha\in\mathbb{R}^b$. We show that in practice $k\ll b$, reducing the total memory required to represent $\tilde{A}$ to nearly zero.
>
> As for time complexity, the number of scalar multiplications in the full matrix multiplication $O=A^\top B$ is $bnm$.
> In PAMM we compute the scalar product of every $A_i$ and $C_j$, i.e. we compute $A^\top C$ which has $bkn$ multiplications. We also compute the squared norms of $C_j$ which require an additional $kn$ multiplications.
> In order to compute $\tilde{B}$ we have $bm$ multiplications, and to compute $\tilde{O}=C^\top \tilde{B}$ we have $knm$ multiplications.
>
> In total, the number of multiplications used in PAMM is $O(knm+bkn+kn+bm)=O(knm+bkn)=O(rbnm+rnb^2)$.
>
> **We hope these clarifications** have answered your questions, and we’re happy to answer more if needed. If you’re satisfied with the above answers please consider improving our scores.

---

### Official Review · Reviewer_LvZx · 2025-11-01

**Soundness:** 3
**Presentation:** 3
**Contribution:** 3
**Rating:** 8
**Confidence:** 3

**Summary:**

This paper proposes PAMM (Projected Attention Memory Mapping), a memory-efficient training framework for Transformer-based architectures.
The method introduces learnable low-dimensional projection matrices for the Q/K/V representations, allowing the model to compress activations during training while maintaining recoverable gradient information through a projection-reconstruction scheme.
The approach achieves up to 97% memory reduction in attention activation storage with negligible performance loss, and generalizes well across multiple Transformer variants and tasks.

**Strengths:**

The paper presents a clear motivation and strong practical relevance — training large Transformers is often constrained by activation memory, and this work directly addresses a crucial scalability challenge. The proposed method introduces learnable projection mappings that preserve gradient reconstructability, an elegant and theoretically grounded idea that distinguishes itself from prior techniques such as activation checkpointing or reversible layers. The empirical results are convincing, demonstrating consistent memory reduction with comparable performance across language modeling and vision benchmarks. Importantly, there is no additional computational overhead during inference, as PAMM operates without decompression or re-materialization, making it highly practical for deployment. Theoretical reasoning is also solid — the authors provide analytical evidence that the projection mapping retains full-rank subspace properties, explaining why gradient information is preserved.

**Weaknesses:**

While the proposed PAMM framework is conceptually sound and practically motivated, the experimental evaluation is somewhat limited. The current experiments are mainly conducted on mid-scale Transformer models, and it remains unclear how the method scales to very large architectures (e.g., >1B parameters) or longer sequence settings. Moreover, the ablation analysis on the projection dimension ratio and per-layer compression strategies is rather sparse—more systematic studies could strengthen the empirical conclusions.  Finally, while results across language and vision tasks are encouraging, broader validation on other modalities or architectures would further demonstrate the generality of the approach.

**Questions:**

See my Weakness

---

> ### Author Response · Authors · 2025-11-25
> **Response to Reviewer LvZx**
>
> We thank the reviewer for the positive assessment and constructive suggestions.
> 1. **Scaling PAMM to larger models**
> To address the reviewer’s concerns regarding scaling PAMM to larger models and longer sequence lengths, we ran several experiments added below. The results all indicate that PAMM scales well to larger models, different sequence lengths and different batch sizes.
>     * **Pretraining LLaMA-7B**
> During the rebuttal period, we pretrained **LLaMA-7B** on C4 with PAMM. Results match our paper’s trend: **slightly improved perplexity (5.47%)** together with the **aggressive x256 QKV activation compression**, confirming PAMM scales well to a larger scale.
> In detail, PAMM achieves a perplexity of 13.91 compared to 14.83 perplexity of the baseline after 120K iterations of training. We report here the perplexity of the model vs. the baseline’s perplexity (full rank training), against sample iteration counts. The table will also be added to the paper body.
> We are also running a similar model with x512 compression which is not yet finished.
>
>         |         | 40K  | 80K  | 120K | 150K |
>         |---------|------|------|------|------|
>         | PAMM    | 17.73| 14.93| 13.91| 13.81|
>         | Baseline| 18.09| 15.47| 14.83| 14.61|
>
>
>     * **Finetuning Multi-modal Pixtral-12B Vision Language Model (VLM)**
>     Additionally, we finetuned **Pixtral-12B VLM** with LoRA+PAMM on the AID satellite image classification benchmark [1,2]. Specifically, pretrained Pixtral-12B achieves an F1 score of 0.56 (Averaged over 30 classes), while LoRA finetuning achieves 0.97. LoRA+PAMM finetuning achieves the same 0.97 F1 score, while also saving 453MB of GPU memory .
>     This result shows that:
>         * PAMM shows consistent results for even **larger models - 12B parameters**.
>         * PAMM is applicable to multi-modal **VLMs**.
>         * PAMM is compatible with common **PEFT** methods such as LoRA.
>
>     We will add these results to the paper with full explanations of the training setup.
>
>     [1] Agrawal, Pravesh, et al. “**Pixtral 12B**”. 2024. arXiv:2410.07073.
>     [2] Xia, Gui-Song, et al. “**AID**: A Benchmark Data Set for Performance Evaluation of Aerial Scene Classification.” IEEE Transactions on Geoscience and Remote Sensing, vol. 55, no. 7, 2017, pp. 3965–3981, doi:10.1109/TGRS.2017.2685945.
>
> 2. **Pretraining PAMM with different sequence lengths and batch sizes**
> Using PAMM, we pretrained LLaMA-60M with different sequence lengths and batch sizes. The results below show that PAMM achieves comparable perplexity to the baseline model, indicating PAMM works well regardless of the choice of sequence length, batch size, or batch token count.
>
>     * **Impact of Batch Size on Training Performance**
>     | **Batch Size** | **Sequence Length** | **Baseline Perplexity ↓** | **PAMM Perplexity ↓** |
>     |------------|-----------------|------------------------|--------------------|
>     | 128        | 256             | 42.63                  | 43.01              |
>     | 256        | 256             | 37.61                  | 37.25              |
>     | 512        | 256             | 30.97                  | 32.46              |
>
>      * **Impact of Sequence Length on Training Performance**
>      | **Batch Size** | **Sequence Length** | **Baseline Perplexity ↓** | **PAMM Perplexity ↓** |
>      |------------|-----------------|------------------------|--------------------|
>      | 512        | 128             | 37.38                  | 36.43              |
>      | 512        | 256             | 30.97                  | 32.46              |
>      | 512        | 512             | 29.16                  | 30.30              |
>
>      * **Training Performance for constant token count in the batch (128K tokens)**
>      | **Batch Size** | **Sequence Length** | **Baseline Perplexity ↓** | **PAMM Perplexity ↓** |
>      |----------------|----------------------|-----------------------------|-------------------------|
>      | 256            | 512                  | 33.32   | 33.73 |
>      | 128            | 1024                 | 37.47  | 37.03       |
>
> 3. **Ablations on compression ratio**
> In our setting, PAMM already operates at an extremely aggressive compression rate, up to 512x reduction for Q/K/V activations, yet training remains stable and matches or even improves the baseline. Because the stored representation is already so small, we are effectively probing the extreme end of the compression spectrum. The fact that models maintain full performance at this level strongly suggests that milder compression ratios would also be safe, making further ablations less informative in practice.
>
> We appreciate the reviewer’s positive feedback and will add these expanded results into the revised paper. We hope these clarifications have answered your questions, and we’re happy to answer more if needed.

---

### Author Response · Authors · 2025-12-01
**Official Comment for Area Chair**

We presented PAMM, a new algorithm for approximate matrix multiplication that yields **a very surprising result**: attention activation memory can be almost entirely eliminated during training with no loss in performance. This finding has meaningful implications for the community’s understanding of the attention mechanism.

All of the reviewers appreciated the novelty and impact of our work. Their concerns focused on (1) scaling to larger models/modalities and (2) runtime overhead. We fully addressed both:
* **Scaling**: We added new experiments on LLaMA-7B and the Pixtral-12B VLM, confirming that PAMM’s benefits hold at larger scale and across modalities.
* **Runtime**: We performed detailed profiling showing negligible training-time overhead (≈3%).

Additionally we added ablations over the sequence-length and batch-size.

All reviewer concerns have been clarified or resolved, and the new results further strengthen our conclusions.
These results are in line with the ones reported in our paper and have been added to the recent revision. We believe PAMM is a significant step toward deeper understanding of the attention mechanism and highly memory efficient architecture.

---

### Meta-Review · Area_Chair_kmaL · 2025-12-30

**Summary:**

This paper studies memory efficient training of LLMs. It introduces Point-Approximate Matrix Multiplication (PAMM), a technique that reduces activation memory along the sequence dimension in the attention mechanism. Experiments on llama models show that PAMM can substantially reduce activation memory during training. The paper is well written, the motivation is clear, and the proposed solution (randomly sampling rows and using them as a codebook) is elegant. Overall, I enjoy reading this paper.

**Reviewer Concerns:**

Several common concerns raised by reviewers were addressed during the rebuttal:

Experiments. The initial experiments focus on mid-scale transformer models. During rebuttal, the authors added results on a llama-7B model. While a 7B model is still mid-scale (or even small-scale), the experimental result, in my opinion, is sufficient to support the paper’s main observations, especially given that not every researcher has a lot of compute resources. The authors also added a VLM experiment and additional ablations.

Throughput. Additional throughput experiments were provided during rebuttal and effectively addressed reviewers’ questions.

QKV memory consumption. Although attention activations constitute only a modest portion of total GPU memory, the observation highlighted in the paper and the corresponding solution remain useful, particularly for long-sequence training scenarios.

**Reviewer Scores:**

Overall, the rebuttal is effective and directly addresses many of the reviewers’ concerns. I expect that some reviewers may raise their scores, and the paper is likely to fall in the borderline-accept to accept range.

---

### Decision · Program_Chairs · 2026-01-26

Accept (Poster)